# Teaching LLMs How to Learn with Contextual Fine-Tuning

Younwoo Choi[1,2*]  Muhammad Adil Asif[1,2*]  Ziwen Han[1†]  John Willes[1,2]
Rahul G. Krishnan[1,2]

[1]University of Toronto  [2]Vector Institute

## Abstract

Prompting Large Language Models (LLMs), or providing context on the expected model of operation, is an effective way to steer the outputs of such models to satisfy human desiderata after they have been trained. But in rapidly evolving domains, there is often need to fine-tune LLMs to improve either the kind of knowledge in their memory or their abilities to perform open ended reasoning in new domains. When human's learn new concepts, we often do so by linking the new material that we are studying to concepts we have already learned before. To that end, we ask, can prompting help us *teach LLMs how to learn.* In this work, we study a novel generalization of instruction tuning, called *contextual fine-tuning*, to fine-tune LLMs. Our method leverages prompts designed to mimic human cognitive strategies in learning and problem-solving to guide the learning process during training, aiming to improve the model's interpretation and understanding of domain-specific knowledge. We empirically demonstrate that this simple yet effective modification improves the ability of LLMs to be fine-tuned rapidly on new datasets both within the medical and financial domains. Project page: https://younwoochoi.github.io/cft-iclr/

## 1 Introduction

Large Language Models (LLMs) have demonstrated strong performance across a wide range of tasks (OpenAI et al., 2024). As model sizes increase, the performance of LLMs on multi-step reasoning, instruction following, and program execution (Wei et al., 2022b; Du et al., 2024) have been found to improve empirically. LLMs are trained via a three-step process: pretraining (which results in a base model) to compress knowledge over a vast text corpus, supervised fine-tuning for instruction following, and alignment with human expectations of behavior for use as a chat bot (Ouyang et al., 2022; Lee et al., 2023; Rafailov et al., 2024; Ethayarajh et al., 2024).

However, LLMs remain unaware of information and events occurring after their knowledge cutoff; in fast-moving domains and in scenarios where deployment requires knowledge of up-to-date information, there is a need to remedy this limitation. There are two popular approaches to this problem. The first is to increase the context length of the model until all anticipated new information fits within this context (the largest of which is Google's Gemini-1.5 model (Reid et al., 2024) with a context length of two million tokens). However, even context lengths this large can be exhausted and it is unclear whether the model's attention mechanism is capable of accurately inferring signal regardless of where it is in the context. The alternate approach uses external knowledge stores via retrieval augmented systems (Lewis et al., 2021). This approach works well when the reasoning abilities already learned by the model suffice to process and extract the relevant information. But gradient-based learning remains vital in scenarios where there is a need to teach the model how to manipulate new tools or learn new strategies for reasoning.

The simplest approach to update model knowledge via fine-tuning is to continue pretraining the base model. Unfortunately, the data and training procedure necessary to replicate the additional fine-tuning and alignment phases are rarely open-sourced in chat-based models. The general practice is to fine-tune the aligned model with new domain-specific knowledge. Models that have undergone

---

*Equal contribution.
†Work conducted while at the Vector Institute

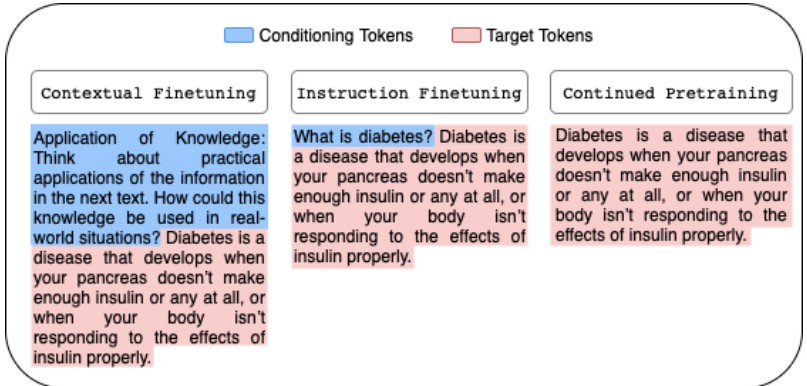

Figure 1: The figure illustrates the distinct approaches of Contextual Fine-Tuning (CFT), Instruction Fine-Tuning (IFT), and Continued Pretraining (CPT). In CFT, a contextual prompt is highlighted in green *"Think about practical applications of the information in the next text. How could this knowledge be used in real-world situations?"* followed by the main text. IFT employs a direct instruction *"What is diabetes?"* before presenting the same text. In contrast, CPT displays only the main text without any preceding prompts or instructions. The key difference lies in CFT's use of contextual prompts that guide the model's semantic understanding and reasoning, whereas IFT relies on explicit instructions to elicit specific responses. CPT, lacking both prompts and instructions, focuses solely on processing the main content.

instruction fine-tuning and alignment training are more amenable to interacting with users but are harder to update with new knowledge. But training to update the knowledge can result in catastrophic forgetting of knowledge gained during pretraining, or loss of capabilities like instruction-following and task-solving (Wang et al., 2023a). There is a need to develop methods that enable us to quickly improve reasoning and recall in aligned LLMs for domain specific fine-tuning.

Our approach is inspired by the capabilities of LLMs to leverage prompts in question answering. For example, few-shot prompting popularized by Brown et al. (2020) performs well on a variety of unseen tasks at prediction time. Wei et al. (2022c) investigated how chain-of-thought (CoT) prompting can significantly improve a model's ability to perform complex multi-step reasoning. Wang et al. (2023b) further improved on CoT by selecting the most consistent answer from a diverse set of sampled reasoning paths.

Our work investigates a simple question: *can prompting improve the efficacy of LLM fine-tuning?* We argue yes, and to this end, we propose a new method for fine-tuning that blends in-context learning with gradient-based learning. In summary, our contributions are as follows:

(a) We present *contextual fine-tuning*, a generalization of instruction tuning, that combines in-context learning and fine-tuning. We further investigate the gradients provided by the additional context and provide synthetic experiments demonstrating their effectiveness for fine-tuning.

(b) To study the impact of our method, we create two datasets in the biomedical domain: the first consisting of 121,489 journal articles from 37 diverse topics in biology and medicine, and second comprising 29 open-source medical textbooks.

(c) We show that contextual fine-tuning can be used to update a model's knowledge more efficiently than continued pretraining and instruction tuning. We show increased performance on both real-world dataset and Q&A tasks while using carefully constructed synthetic data to better understand where performance gains arise from.

## 2 RELATED WORK

**Instruction Tuning** The common paradigm used in training instruction aligned ("chat") LLMs involves three steps: pretraining on unlabeled corpora, performing instruction tuning, followed by reward-based preference training, as used in Ouyang et al. (2022). The instruction tuning phase is generally used as a first step to aligning LLMs to human instructions or when access to human-labeled preference data is limited. Instruction tuning significantly narrows the divide between models' traditional next-word prediction objectives and the practical need for models to adhere to explicit

human instructions. Wei et al. (2022a) have highlighted how this approach markedly boosts zero-shot performance across previously unseen tasks, underlining its effectiveness. Earlier work (Wei et al., 2022a; Iyer et al., 2022) have proposed using a large set of instructions for NLP tasks, while more recent findings (Wang et al., 2023c; Taori et al., 2023; Peng et al., 2023; Zhou et al., 2024) have found success using increasingly smaller and higher quality instruction datasets on open sourced pretrained models such as Llama (Touvron et al., 2023a). Notably, as Gudibande et al. (2023) discovers, training on instructions in the instruction tuning phase does not improve the underlying capabilities of the models; these models merely imitate the instruction following template. Recent developments have explored alternative approaches to standard instruction tuning. Wang et al. (2025) propose Critique Fine-Tuning, where models learn to critique noisy responses rather than imitate correct ones, demonstrating improvements over SFT across mathematical reasoning benchmarks.

The primary distinction between contextual fine-tuning and instruction fine-tuning lies in the semantic content of the tokens used as input. Instruction tuning has input-output pairs $(x, y)$ that are data point specific (e.g., $x$ = "Who is the current president of the United States?", $y$ = "Joe Biden"). Within instruction tuning, there is a specific, narrow question for which there exists a *right* answer that the model is expected to identify. In contextual fine-tuning, our intent is to pair $y$ with a randomly sampled contextual prompt $x$ which serves as guidance for the model to learn the most important information. $x$ can be specific or general desiderata useful for learning intended to prime the model to contextualize and incorporate the knowledge in $y$ within its parameters.

**Domain-Specific Training** While pretraining on trillions of unlabeled tokens creates generalist foundation models (Bommasani et al., 2021; Touvron et al., 2023a;b; Anil et al., 2023; OpenAI et al., 2024), injecting domain-specific expertise into models while retaining the generalist remains an active front of research. Improving the underlying capabilities of LLMs is a more difficult challenge than simply aligning to instructions, in part due to the much larger dataset requirement. While smaller LLMs are capable of outperforming the larger monoliths in specific domains (Gunasekar et al., 2023; Li et al., 2023), or pushing language modeling in a simplified domain to the extreme (Eldan & Li, 2023). Several lines of work have explored using better datasets to improve performance on existing pretrained models as a form of continual learning in areas such as medicine, finance, chip design, coding, and mathematics (Chen et al., 2023b; Wu et al., 2023; Liu et al., 2023; Rozière et al., 2024; Paster et al., 2023; Azerbayev et al., 2023). Such continued training runs the risk of model forgetting, in which general reasoning and capabilities in other domains tend to decline (Luo et al., 2023; Wang et al., 2023a; Chen et al., 2023a).

## 3 METHODOLOGY

### 3.1 NOTATION & BACKGROUND

We consider a large language model (LLM) $P_\theta$, parameterized by pretrained weights $\theta$. We have access to a domain-specific corpus $\mathcal{D}_{train}^{raw}$ consisting of sequences of tokens. Our objective is to fine-tune the model to obtain new parameters $\theta'$ that enhance performance on domain-specific downstream tasks, evaluated on a test set $\mathcal{D}_{test}$.

**Continued pretraining.** Continued pretraining (CPT) leverages large volumes of unlabeled domain-specific data to refine the model's understanding of the domain. Given sequences of tokens $x = (x_1, x_2, \ldots, x_n)$ sampled from $\mathcal{D}_{train}^{raw}$, the model is trained using the causal language modeling objective, which predicts the next token given the previous tokens.

$$\mathcal{L}_{CPT}(\theta) = -\mathbb{E}_{x \sim \mathcal{D}_{train}^{raw}} \sum_{k}^{n} \log P_\theta(x_k \mid x_{<k}). \tag{1}$$

where $x_{<k} = (x_1, x_2, \ldots, x_{k-1})$ represents the sequence of tokens preceding $x_k$.

**Instruction fine-tuning.** Instruction fine-tuning utilizes a collection of instruction-response pairs $(x, y)$ sampled from a dataset $\mathcal{D}_{train}^{IFT}$. Here, $x$ is an instruction or prompt, and $y$ is the corresponding response. The model is trained to generate the response $y$ conditioned on the instruction $x$.

$$\mathcal{L}_{IFT}(\theta) = -\mathbb{E}_{(x,y) \sim \mathcal{D}_{train}^{IFT}} \sum_{k}^{m} \log P_\theta(y_k \mid x, y_{<k}). \tag{2}$$

where $y = (y_1, y_2, \ldots, y_m)$ and $y_{<k} = (y_1, y_2, \ldots, y_{k-1})$.

## 3.2 CONTEXTUAL FINE-TUNING

We introduce Contextual Fine-tuning (CFT), a method that incorporates contextual prompts into the training process to guide the model's learning in a domain-specific manner. Inspired by constructivist learning theory (Piaget, 1952), which emphasizes active engagement and thoughtful processing for effective learning, we hypothesize that contextual prompts can enhance the model's ability to internalize and reason about new concepts within the domain.

**Designing contextual prompts.** We define a set of contextual prompts $\mathcal{C} = \{c^{(1)}, c^{(2)}, \ldots, c^{(L)}\}$, where each prompt $c^{(l)} = (c_1^{(l)}, c_2^{(l)}, \ldots, c_{n_l}^{(l)})$ is a sequence of tokens for some length $n_l$, designed to guide the model during training. These prompts mimic effective human learning strategies by encouraging the model to engage in various cognitive processes such as focusing on key concepts, critical analysis, and application of knowledge.

*Prompt design from educational strategies:* We select 10 prompts to provide a diverse yet manageable set of learning strategies to balance between offering sufficient variation to cover different cognitive approaches and maintaining practicality in training. We present four of these prompts in the main text and include the remaining six in Appendix A.1. Each prompt is grounded in established educational theories, as detailed below:

1. Focus on Key Concepts: This prompt aligns with Sweller (2011), which emphasizes the importance of reducing unnecessary cognitive load to facilitate learning. By focusing on essential information, learners can allocate their cognitive resources more effectively.

   - *"Concentrate on understanding the core principles and essential facts in the following text. Pay special attention to definitions, examples, and conclusions."*

2. Contextual Understanding: Piaget (1952) suggests that learners build new knowledge upon the foundation of their existing understanding by making connections between new and prior information.

   - *"As you read the next passage, relate its content to its broader context and implications. Think about how this information connects to what you've learned previously."*

3. Critical Analysis: This prompt is supported by Bloom et al. (1956), which encourages higher-order thinking skills such as analysis, evaluation, and synthesis, essential for deep learning and understanding.

   - *"Critically analyze the upcoming information. Look for underlying assumptions, evaluate arguments, and consider different perspectives."*

4. Question-Based Learning: Paul & Elder (2006) promote critical thinking by encouraging learners to engage with the material through probing questions, leading to deeper comprehension.

   - *"Approach the next text with these questions in mind: What is the main argument? How is evidence used to support it? What are the implications of these findings?"*

The selection of prompts inspired by educational theories satisfy two important criteria – they are compact and general-purpose (in that they are applicable across different domains). Indeed, these prompts need not necessarily be *optimal*, rather they need only contain semantic content effective for modifying the gradients during the process of fine-tuning.

*Text-adaptive contextual prompts:* As an alternative to manually designed prompts, we study contextual prompts generated by instructing GPT-4o-mini (OpenAI, 2024). The LLM was tasked to create prompts depending on the content of each training batch. This adaptive approach serves as an alternative to using fixed prompts derived from educational strategies, allowing us to study the value of adaptive prompts generated by another LLM. For example, GPT4o-mini generated the following prompt: *'Critically evaluate the methodologies and findings presented in this study on PCR techniques and LeHV-5 detection. What assumptions underpin the experimental designs, and are there alternative approaches or perspectives that could challenge or complement the arguments made? Consider the implications of these methodologies for broader scientific research and diagnostics in veterinary medicine.'* See Appendix D.3 for additional details on the methodology of this approach.

**Learning with contextual prompts.** For each training example, we integrate a contextual prompt to guide the model's focus. The procedure is as follows:

1. **Sampling:** Given a domain-specific text sequence $x = (x_1, x_2, \ldots, x_n)$ sampled from $\mathcal{D}_{train}^{raw}$ and we randomly select a contextual prompt $c$ from $\mathcal{C}$.

2. **Input Construction:** We prepend the prompt $c = (c_1, c_2, \ldots, c_m)$ to the text sequence $x$ to form the new input sequence: $x' = (c_1, c_2, \ldots, c_m, x_1, x_2, \ldots, x_n)$.

3. **Training Objective:** The model is trained to predict the tokens in $x$, conditioned on both the prompt $c$ and the preceding tokens in $x$. The loss function for CFT is defined as:

$$\mathcal{L}_{CFT}(\theta) = -\mathbb{E}_{x \sim \mathcal{D}_{train}^{raw}, c \sim \mathcal{C}} \sum_{k=1}^{n} \log P_\theta(x_k \mid c, x_{<k}). \tag{3}$$

Refer to Algorithm 1 in Appendix A.2 for the detailed algorithm. We hypothesize that by incorporating contextual prompts during training, we can influence the model's learning trajectory, aligning the gradients towards more semantically meaningful representations. These gradients guide the optimization process, encouraging the model to develop a deeper understanding of the content.

## 4 OPENMEDTEXT

To evaluate the effectiveness of contextual fine-tuning in a domain-adaptive setting, we curated a dataset consisting of both academic journal articles and educational textbooks. Our objective was to assemble a corpus that not only covers a wide range of topics within bio-medicine but also provides structured textual data (of varying levels of quality) suitable to align with our goal of studying how well LLMs can learn using contextual prompts. The inclusion of textbooks provides structured and pedagogically organized content, which is conducive to the learning processes we aim to emulate.

The rationale for going for quantity rather than highly curated quality in the data we collected was to have a realistic representation of internet scale data (albeit within a constrained domain) and to showcase how contextual fine-tuning could improve learning *even-if* the data was of mixed quality.

Our dataset differs from existing biomedical corpora such as PubMed Central (PMC) in several ways. While PMC provides a vast collection of biomedical literature, it predominantly consists of research articles focused on specific studies and often lacks the pedagogical structure found in textbooks. In contrast, our dataset integrates both detailed research articles and educational textbooks, offering a combination of depth and structured learning materials. This integration makes it a valuable testing ground to assess the viability of prompts that leverage understanding past relationships when absorbing future ones.

**MDPI Journals:** We collected 121,489 biomedical journal articles from MDPI, covering 37 diverse topics such as antibiotics, biomedicines, diseases, and cardiology (see Table 6 and 7 for detailed lists). The selection of MDPI journals was motivated by their open-access policy and the breadth of biomedical subjects they cover, ensuring a wide-ranging representation of biomedical research.

**Medical Textbooks:** We also curated 29 open-source medical textbooks into our dataset. Textbooks were chosen because they provide structured, comprehensive overviews of medical knowledge, organized pedagogically to facilitate learning. This aligns with our objective of leveraging contextual fine-tuning to enhance the learning processes of language models, as textbooks inherently contain explanations, definitions, and educational narratives beneficial for model training. The data we collect have the following characteristics:

1. **Coverage**: The dataset incorporates a wide array of topics derived from both medical journals and textbooks, ensuring extensive coverage of biomedicine. Unlike existing datasets such as PubMed, which primarily consist of research articles and abstracts, our dataset combines journals with the structured educational content of textbooks.

2. **Alignment with Educational Objectives**: The inclusion of textbooks provides structured and pedagogically organized material, which is particularly suitable for our contextual fine-tuning approach. Textbooks facilitate a learning process analogous to human education, supporting the models in acquiring and retaining biomedical concepts effectively.

3. **Quality of text tokens**: We have meticulously cleaned and pre-processed the texts to remove irrelevant sections and ensure clarity. This cleaning process reduces noise and potential sources of error, enhancing the quality of the data, which in turn improves the accuracy and reliability of models trained using this dataset.

## 5 CONTEXTUAL FINE-TUNING EXPERIMENTS

### 5.1 EXPERIMENTAL SETUP

We assess the efficacy of contextual fine-tuning by comparing the performance of large language models (LLMs) when fine-tuned on domain-specific corpora using both contextual fine-tuning and standard unsupervised fine-tuning approaches. The performance of LLMs is measured using the relevant downstream tasks.

**Datasets.** We evaluate the effectiveness of contextual fine-tuning across two distinct domains: the financial domain and the biomedical domain. For the financial domain, we use a dataset comprising 306,242 financial news articles (Jeet, 2018). In the biomedical domain, we utilize OpenMedText, as described in detail in the previous section. When incorporating instruction fine-tuning into our experiments, we include additional datasets specific to each domain. For the financial domain, we use FinAlpaca (Gaurang Bharti, 2024), which contains instruction-output pairs tailored for financial tasks. In the biomedical domain, we supplement with datasets from (OpenGPT, 2023), providing question-answer pairs bootstrapped from the NHS encyclopedia (NHS UK, 2023). Additionally, we incorporate UltraChat (Ding et al., 2023), a large-scale, multi-round dialogue dataset, into our instruction fine-tuning process.

**Benchmarks.** The effectiveness of the fine-tuning approach in each domain is evaluated using several domain-specific benchmarks. In the financial domain, we consider (1) the sentiment analysis task **FiQA** (Xie et al., 2023) where LLMs predict sentiments categorized as 'positive', 'neutral', or 'negative' in financial texts. (2) The headline classification task **MultiFin** (Jørgensen et al., 2023; Xie et al., 2023), where LLMs categorize each news article into one of six categories based on the headline. (3) **Causal20** (Xie et al., 2023), which involves classifying sentences extracted from financial news as either depicting a 'causal' or 'noise' relationship between financial events. For the biomedical domain, we consider the following multiple-choice question (MCQ) datasets from **Massive Multitask Language Understanding** (MMLU) (Hendrycks et al., 2021): (1) Anatomy, (2) Clinical Knowledge, (3) College Biology, (4) College Medicine, (5) Medical Genetics, and (6) Professional Medicine. We also use **MedQA**, a collection of multiple-choice questions from the professional medical board exams (Jin et al., 2020).

**Training and evaluation details.** We use several configurations of the Llama-2 models to evaluate the effectiveness of contextual fine-tuning. More details can be found under Appendix A.3. In our evaluation, MCQs are formatted with questions followed by several options labeled with ID symbols (e.g., `A`/`B`/`C`/`D`). Building on the approach outlined in Zheng et al. (2024), we instruct the language models to predict an option ID symbol rather than the textual content of the answer. This method addresses a critical issue: the likelihood of the answer's text being naturally plausible could be conflated with its likelihood of being the correct response due to the model's linguistic biases. However Robinson & Wingate (2023) raises concerns regarding LLMs' inherent selection biases, which highlights that these models may show a preference for specific option IDs. To counteract this bias and enhance the validity of our evaluations, we adopt one of debiasing techniques as prescribed in the aforementioned work. See Appendix C.1 for the detailed method.

### 5.2 RESULTS

We evaluate the zero-shot performance of large language models (LLMs) trained with different methods across both medical and financial benchmarks. See Appendix E.3 for extended results on general-purpose and instruction-following benchmarks. It is important to note that our primary objective is not to achieve state-of-the-art performance but to assess the relative improvements offered by contextual fine-tuning (CFT), instruction fine-tuning (IFT), and continued pretraining (CPT) compared to a baseline model. We focus primarily on a metric for evaluating the effectiveness of these training approaches: $\%\Delta_{CPT}^{CFT}$, which denotes the performance difference between CFT and CPT.

**Contextual fine-tuning is effective across model scales.** We asses how contextual fine-tuning performs across different model scales. Table 1 presents medical benchmarks for the 7B and 13B

| | Accuracy (↑) | | | | | | | |
|---|---|---|---|---|---|---|---|---|
| **Llama 2 7B** | Anatomy | Clinical Knowledge | College Biology | College Medicine | Medical Genetics | Professional Medicine | MedQA | Average |
| Chat | 44.07 | 46.79 | 48.61 | 39.02 | 49.00 | **48.90** | 38.96 | 45.05 |
| Chat (CPT) | 45.19 | 47.17 | 49.31 | 43.93 | 50.50 | 46.32 | 39.28 | 45.96 |
| Chat (CFT) | **48.15** | **48.87** | **52.08** | **44.22** | **54.00** | 46.69 | **40.65** | **47.81** |
| | | | | | | | | |
| **Llama 2 13B** | Anatomy | Clinical Knowledge | College Biology | College Medicine | Medical Genetics | Professional Medicine | MedQA | Average |
| Chat | 51.85 | 56.60 | 54.17 | 46.82 | **63.50** | 56.99 | **45.33** | 53.61 |
| Chat (CPT) | 50.37 | 60.00 | 55.90 | 50.58 | 62.00 | 57.35 | 43.95 | 54.31 |
| Chat (CFT) | **53.33** | **63.21** | **57.99** | **56.35** | 62.50 | **57.72** | 44.85 | **56.56** |

Table 1: Medical Benchmarks (Zero-shot). The results show that the 7B model achieved a $\%\Delta^{CFT}_{CPT}$ of 1.85%. The 13B model demonstrated increased effectiveness with a $\%\Delta^{CFT}_{CPT}$ of 2.25%, indicating that CFT's impact grows with the model's scale.

| | FiQA | Causal 20 | Multifin | |
|---|---|---|---|---|
| **Llama 2 7B** | F1 | F1 | F1 | Average |
| Chat | 56.40 | **90.40** | 38.74 | 61.48 |
| Chat (CPT) | 62.53 | 90.16 | 38.23 | 63.64 |
| Chat (CFT) | **67.69** | 90.17 | **46.01** | **67.96** |

Table 2: Llama 2 7B Financial Benchmarks (Zero-shot).

| | FiQA | Causal 20 | Multifin | |
|---|---|---|---|---|
| **Llama 2 13B** | F1 | F1 | F1 | Average |
| Chat | 61.18 | 84.77 | 45.81 | 63.92 |
| Chat (CPT) | 66.96 | **90.06** | 45.33 | 67.45 |
| Chat (CFT) | **70.55** | 89.87 | **50.94** | **70.45** |

Table 3: Llama 2 13B Financial Benchmarks (Zero-shot).

model scales which shows a $\%\Delta^{CFT}_{CPT} = 1.85\%$. For the 13B model, these metrics increase to $\%\Delta^{CFT}_{CPT} = 2.25\%$. Similarly, Tables 2 and 3 contain financial benchmarks. The 7B model records a $\%\Delta^{CFT}_{CPT} = 4.32\%$ whereas the 13B model, the results are $\%\Delta^{CFT}_{CPT} = 3\%$. The results demonstrate that the simple augmentation of contextual prompting can help increase performance across the board.

**Contextual fine-tuning is preferable to existing approaches for improving a model at a fixed scale.** The tables in Appendix D.1 show the performance on the medical and financial benchmarks while holding the model scale constant. The base non-instruct model holds an average accuracy of 41.34% on the medical benchmarks. Our experiments find that combining training schemes provides the greatest boost in fine-tuning performance. In particular, combining CFT and IFT gives a performance boost of $\%\Delta^{CFT+IFT}_{Base} = 2.95\%$ compared to $\%\Delta^{CPT+IFT}_{Base} = 1.91\%$. Similar trends are seen in the financial benchmarks where the same combination led to an increase of $\%\Delta^{CFT+IFT}_{Base} = 36.28\%$ in F1 score. These results further solidify that augmenting the CPT stage of fine-tuning to instead use CFT provides a near-free boost in performance. More detailed analyses can be found in Appendix D.1. Additionally, we compare CFT with AdaptLLM (Cheng et al., 2024), a method of domain-specific continued pretraining on medical benchmarks. Table 4 shows that CFT consistently achieves higher accuracy across all tasks by 4.89% on average. This result highlights CFT's superior effectiveness on our full-text medical datasets compared to AdaptLLM. Refer to Appendix E.2 for more details on the AdaptLLM methodology.

**The semantic content of the contextual prompts are important to improving performance.** The core aspect of our study involves examining the impact of additional context on model performance, and specifically how the signal from this context provides a boost in learning performance. We conduct an ablation by introducing negative contextual prompts, which are designed to mislead the model by suggesting that the following information is incorrect. These results can be found in tables under Appendix D.2. In the financial domain, the impact of negative prompts is evident. The 7B model experienced a performance drop of $\%\Delta^{-CFT}_{CFT} = -3.41\%$, and the 13B model sees a decrease of $\%\Delta^{-CFT}_{CFT} = -2.39\%$. All models undergoing negative contextual fine-tuning still perform better than those subjected to CPT. In addition to experiments with negative contextual prompts, we investigate text-adaptive Contextual Fine-Tuning (TextAdaptCFT) using automatically generated, text-dependent prompts. As shown in Table 5, TextAdaptCFT achieves an average accuracy of 46.31%, outperforming both the baseline Chat model (45.05%) and the Chat CPT model (45.96%). This suggests that our manually constructed contextual prompts are not the only viable solution, and that semantic content within prompts plays a crucial role in enhancing model performance. Refer to Appendix D.3 for additional details on our automated prompt construction process.

| | Accuracy (↑) | | | | | | | |
|---|---|---|---|---|---|---|---|---|
| Llama 2 7B | Anatomy | Clinical Knowledge | College Biology | College Medicine | Medical Genetics | Professional Medicine | MedQA | Average |
| Chat | 44.07 | 46.79 | 48.61 | 39.02 | 49.00 | **48.90** | 38.96 | 45.05 |
| Chat (CPT) | 45.19 | 47.17 | 49.31 | 43.93 | 50.50 | 46.32 | 39.28 | 45.96 |
| Chat (CFT) | **48.15** | **48.87** | **52.08** | 44.22 | **54.00** | 46.69 | **40.65** | **47.81** |
| AdaptLLM | 44.45 | 47.36 | 48.27 | 39.60 | 45.00 | 38.61 | 37.12 | 42.92 |

Table 4: Medical Benchmarks (Zero-shot). The results show that the 7B model trained with CFT outperforms the model trained with AdaptLLM by 4.89% on average.

| | Accuracy (↑) | | | | | | | |
|---|---|---|---|---|---|---|---|---|
| **Llama 2 7B** | Anatomy | Clinical Knowledge | College Biology | College Medicine | Medical Genetics | Professional Medicine | MedQA | Average |
| Chat | 44.07 | 46.79 | 48.61 | 39.02 | 49.00 | **48.90** | 38.96 | 45.05 |
| Chat (CPT) | 45.19 | 47.17 | 49.31 | 43.93 | 50.50 | 46.32 | 39.28 | 45.96 |
| Chat (CFT) | **48.15** | **48.87** | **52.08** | 44.22 | **54.00** | 46.69 | **40.65** | **47.81** |
| Chat (TextAdaptCFT) | 45.56 | 48.12 | 49.31 | **44.80** | 52.50 | 43.57 | 40.34 | 46.31 |

Table 5: Medical Benchmarks (Zero-shot). TextAdaptCFT outperforms the baseline Chat model and the Chat CPT model by 1.26% and 0.35% on average, respectively.

## 5.3 SYNTHETIC EXPERIMENTS

We hypothesize that the effectiveness of CFT stems from changing the gradients obtained via conditioning on prompts. These serve to regularize the learning process during fine-tuning. Testing this hypothesis directly is challenging since (a) different LLMs might interpret semantic information in a prompt differently and (b) it requires knowing which neurons are responsible for representing the inferred semantic information in the prompt. The primary objective of our synthetic experiments is to analyze how contextual prompts affect the gradients of transformer models during training in a simplified controlled setting where we can describe the semantic information that is necessary for learning explicitly via text.

**Setup.** Using the framework of Garg et al. (2022), we first pre-train a transformer model to learn a class of linear functions $\mathcal{F} = \{f \mid f(x) = w^\top x, w \in \mathbb{R}^d\}$, where $w \sim \mathcal{N}(0, I_d)$ and $x \sim \mathcal{N}(0, I_d)$. We then investigate how different fine-tuning strategies affect the model's ability to learn a new compositional function class $\mathcal{G}$ where each $g \in \mathcal{G}$ is a composition $g(x) = h(f(x))$ for some $f \in \mathcal{F}$. This setup models real-world scenarios where a pretrained LLM understands basic patterns (represented by $f$), and needs to quickly learn new tasks (represented by $h$) that build upon this understanding. We explore two distinct compositional structures: polynomial combinations $g(x) = f(x) + f(x)^2$ and multiple linear relationships $g(x) = f(x) + w_2^\top x$. For each structure, we evaluate three fine-tuning strategies with that differ in their prompt construction $P$:

1. *Continued Pretraining (CPT)*: $P_{CPT} = (x_1, g(x_1), x_2, g(x_2), \ldots, x_k, g(x_k))$ only exposes the model to inputs and their corresponding outputs under $g(x)$, without explicitly providing $f(x)$.

2. *Contextual Fine-Tuning (CFT)*: $P_{\text{CFT}} = (x_1, f(x_1), \ldots, x_k, f(x_k), x_1, g(x_1), \ldots, x_k, g(x_k))$ provides explicit access to both $f(x)$ and $g(x)$ during fine-tuning, allowing the model to learn the relationship between them.

3. *Negative Contextual Fine-Tuning (NEG-CFT)*: $P_{\text{NEG-CFT}} = (x_1, r_1, \ldots, x_k, r_k, x_1, g(x_1), \ldots, x_k, g(x_k))$, where $r_i \sim \mathcal{U}(0, 1)$. This is an ablation where random values replace $f(x)$ in the context, providing non-helpful information.

This construction allows us to directly measure how different training strategies affect the model's ability to efficiently learn compositional functions. We refer the reader to Appendix F for the full experiment breakdown.

**Contextual fine-tuning improves learning dynamics.** Figure 4a and 4c illustrate that transformers fine-tuned using CFT achieve lower loss compared to those trained with CPT and NEG-CFT suggesting that the content of the contextual prompts in both cases better guides training dynamics when learning a new function class.

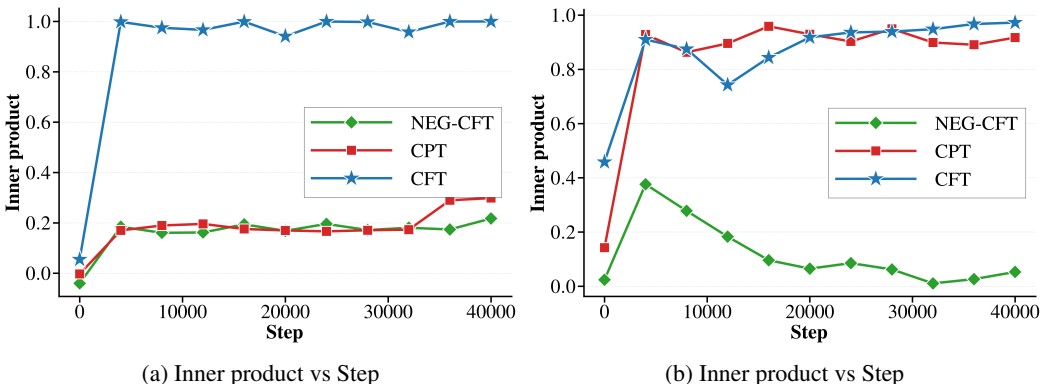

(a) Inner product vs Step          (b) Inner product vs Step

Figure 2: We examine the gradients during Contextual Fine-Tuning (CFT), Continued Pretraining (CPT), and Negative Contextual Fine-Tuning (NEG-CFT) on learning new function classes—polynomial combination (a) and multiple linear relationships (b). (a) and (b) illustrate the normalized inner product between the transformer's gradients and the true gradients $\nabla_x g(x_{query})$, where CFT exhibits a higher alignment, approaching 1, indicating effective learning of the target functions.

**Contextual prompts help the model capture the underlying functional relationships.** We examine the gradients of the transformer across different training strategies. We look at the case where the transformer's input is of the form $P = (x_1, f(x_1), \ldots, x_k, f(x_k), x_{query})$ and its output aims to approximate $f(x_{query})$. Consequently, the gradient of the transformer's output with respect to $x_{query}$ should align with the gradient $\nabla_x g(x_{query})$. In our experiments, we compute the normalized inner product between the gradient of the transformer's output and the true gradient $\nabla_x g(x_{query})$ during training. For the polynomial combination class $\mathcal{G}$, the gradient is:

$$\nabla_x g(x_{query}) = w_1 + 2(w_2^\top x_{query}) w_2 \tag{4}$$

and for the multiple linear relationships class,

$$\nabla_x g(x_{query}) = w_1 + w_2 \tag{5}$$

Figure 2a demonstrates that the gradients from the CFT-trained transformer exhibit a much higher alignment with $\nabla_x g(x_{query})$ compared to those from CPT and NEG-CFT. The inner product between the gradients approaches 1 for CFT, indicating near-perfect alignment. This close alignment suggests that the model's updates are effectively moving in the direction that minimizes the loss concerning the target function $g$. Essentially, the transformer not only predicts $g(x)$ accurately but also captures the underlying functional relationships due to the informative contextual prompts. Figure 2b highlights the importance of the content within the contextual prompts. Despite NEG-CFT having a similar prompt structure to CFT, the use of random or non-informative values in place of $f(x_i)$ results in gradients that do not align well with $\nabla_x g(x_{query})$. This misalignment indicates that the relevance and quality of the content of contextual prompts are crucial for guiding the model's learning process effectively.

## 6   CONCLUSION

This study introduces contextual fine-tuning, a variation of instruction tuning, which leverages contextual gradients to guide the learning process through simple, domain-adaptive prompts. Our experiments reveal that the contextual gradients enhance performance by effectively direct model learning, demonstrating superior results over traditional continued domain pretraining in both financial and medical domains. Finally, we open-source a biomedical dataset curated from MDPI journals and other open-source medical textbooks.

Despite these promising results, there remains more to do to better understand this phenomena. We conjecture that the effectiveness of our method is related to explanatory text present in the training corpora. Prystawski et al. (2023) suggests that chain-of-thought prompting works because local reasoning steps embedded in pretraining corpora simulate sequences of steps that lead to conclusion which are relied on at test time. We hypothesize that similar such contextual cues exist in pretraining data. Future studies examining this carefully would be valuable to test this conjecture and shed light

on the mechanisms by which prompts provide useful supervisory signals during learning. Additionally, we have experimented with CFT with data that likely contain a high density of information such as medical journals and textbooks. This rich semantic content allows contextual prompts to effectively guide the learning process by influencing gradient updates. In future work, it would be valuable to apply contextual fine-tuning to different types of data, including those with longer context lengths or lower information density, such as Reddit posts.

## 7 ACKNOWLEDGMENTS

We acknowledge Aryan Dhar for early implementations of tools to assist with creating the textbook datasets. RGK is supported by a Canada CIFAR AI Chair. This research was supported by an Amazon Research Grant and an NFRF Special Call NFRFR2022-00526. Resources used in preparing this research were provided, in part, by the Province of Ontario, the Government of Canada through CIFAR, and companies sponsoring the Vector Institute.

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

# Contents

# A  CONTEXTUAL FINE-TUNING DETAILS

## A.1  CONTEXTUAL PROMPTS

The full list of contextual prompts is provided below:

1. Application of Knowledge: Grounded in Situated Learning Theory (Lave & Wenger, 1991), this prompt emphasizes that learning is most effective when contextualized and applied in real-world scenarios. Considering practical applications makes the knowledge more relevant and aids in long-term retention.

   - *"Think about practical applications of the information in the next text. How could this knowledge be used in real-world situations?"*

2. In-Depth Exploration: Craik & Lockhart (1972) indicates that deeper, more elaborate processing of information leads to better memory retention compared to shallow processing.

   - *"Dive deep into the details and nuances of the following content. Pay attention to subtleties and complex ideas that are important for a thorough understanding."*

3. Reflective Thinking: Informed by Reflective Practice theories (Schon, 1984), this prompt encourages learners to critically reflect on new information and its impact on their existing beliefs. Reflective thinking fosters self-awareness and facilitates continuous learning and personal growth.

   - *"Reflect on the information presented in the next passage. Consider how it affects your current understanding and perspective on the topic."*

4. Creative Interpretation: This prompt promotes Divergent Thinking as part of Guilford's Structure of Intellect model (Guilford, 1967). Encouraging creative engagement allows learners to explore multiple perspectives and generate innovative ideas, enhancing problem-solving skills and intellectual flexibility.

   - *"Engage creatively with the upcoming text. Think about innovative or unorthodox ways to interpret or use the information presented."*

5. Summarization and Synthesis: Wittrock (1974) suggests that learners understand and remember information better when they actively generate relationships and summaries in their own words.

   - *"Summarize the main points of the following content in your own words. Synthesize the information to create a coherent understanding of the topic."*

6. Focus on Key Concepts: This prompt aligns with Sweller (2011), which emphasizes the importance of reducing unnecessary cognitive load to facilitate learning. By focusing on essential information, learners can allocate their cognitive resources more effectively.

   - *"Concentrate on understanding the core principles and essential facts in the following text. Pay special attention to definitions, examples, and conclusions."*

7. Contextual Understanding: Piaget (1952) suggests that learners build new knowledge upon the foundation of their existing understanding by making connections between new and prior information.

   - *"As you read the next passage, relate its content to its broader context and implications. Think about how this information connects to what you've learned previously."*

8. Critical Analysis: This prompt is supported by Bloom et al. (1956), which encourages higher-order thinking skills such as analysis, evaluation, and synthesis, essential for deep learning and understanding.

   - *"Critically analyze the upcoming information. Look for underlying assumptions, evaluate arguments, and consider different perspectives."*

9. Question-Based Learning: Paul & Elder (2006) promote critical thinking by encouraging learners to engage with the material through probing questions, leading to deeper comprehension.

- *"Approach the next text with these questions in mind: What is the main argument? How is evidence used to support it? What are the implications of these findings?"*

10. Comparative Learning: Based on Relational Frame Theory (Steven C. Hayes, 2001), this prompt enhances understanding by encouraging learners to relate new information to existing knowledge structures.

  - *"Compare and contrast the upcoming information with what you have learned in similar topics. Look for differences, similarities, and connections."*

## A.2 ALGORITHM

---

**Algorithm 1** Contextual Fine-tuning (CFT)

---

**Require:** pretrained model $P_\theta$, domain-specific corpus $\mathcal{D}_{\text{train}}^{\text{raw}}$, set of contextual prompts $\mathcal{C}$, batch size $B$

1: **for** each training step **do**
2:     Sample a batch of texts $\{x^{(i)}\}_{i=1}^B$ from $\mathcal{D}_{\text{train}}^{\text{raw}}$
3:     **for** each text $x^{(i)}$ in the batch **do**
4:         Sample a contextual prompt $c^{(i)}$ uniformly at random from $\mathcal{C}$
5:         Construct the input sequence $x'^{(i)} = (c^{(i)}, x^{(i)})$
6:         Set the target sequence $y^{(i)} = x^{(i)}$
7:     **end for**
8:     Compute the loss:

$$\mathcal{L}(\theta) = -\frac{1}{B} \sum_{i=1}^B \sum_{k=1}^{n^{(i)}} \log P_\theta \left( y_k^{(i)} \mid c^{(i)}, x_{<k}^{(i)} \right)$$

9:     Update model parameters $\theta$ using gradients $\nabla_\theta \mathcal{L}(\theta)$
10: **end for**

---

## A.3 TRAINING DETAILS

In our experimental setup, we use several configurations of the Llama-2 models to evaluate the effectiveness of contextual fine-tuning compared to standard unsupervised fine-tuning. Specifically, we employ the Llama-2 Base model with 7 billion parameters and Llama-2 Chat models with both 7 billion and 13 billion parameters, each with a sequence length of 4096. All models are both contextual and unsupervised fine-tuned. For contextual fine-tuning, we employ Equation 3. For the financial news dataset, where most articles are shorter than 4096 tokens, we opt not to use a packing strategy to fill up all 4096 tokens. Instead, we pad any remaining space. This ensures that each semantically distinct text is associated with its own contextual prompt. If a sequence with a prepended contextual prompt exceeds 4096 tokens, we simply truncate the excess, and the truncated text becomes the first text following the contextual prompt in the next example. The models are trained for one epoch with a batch size of 128 and a learning rate of 2e-5. To assess the efficiency of CFT, we carefully measured the computational resources required for our experiments and compared the overhead introduced by incorporating contextual prompts. Below are the details of our computational setup and findings. We utilized the Fully Sharded Data Parallel (FSDP) training to efficiently distribute the model across multiple GPUs. Training was performed using the bf16 (Brain Floating Point) data format. We implemented Flash Attention 2 (Dao, 2024). All training was conducted with 8 NVIDIA A100 GPUs. With the above configuration, we achieved a training speed of approximately 55,188 tokens per second, measured using the Llama tokenizer. The fine-tuning required a total of approximately 111.11 GPU-hours to complete. Incorporating contextual prompts increased the total training time by approximately 0.89 GPU-hours, resulting in a total of 112 GPU-hours. Each contextual prompt added only about 0.8% to the length of each training example on average. This slight increase in input length led to less than a 1% increase in total training time.

## B OPENMEDTEXT

### B.1 MDPI JOURNALS DETAILS

See Table 6 for the full token breakdown.

| Journal Category | Number of Journals | Number of Tokens |
|---|---|---|
| Allergies | 25 | 140,865 |
| Antibiotics | 3604 | 26,572,807 |
| Antibodies | 380 | 3,248,253 |
| Behavioral Science | 1047 | 7,553,809 |
| Biologics | 32 | 261,410 |
| Biomedicines | 3909 | 34,783,559 |
| Biomedical Informatics | 26 | 201,656 |
| Biomolecules | 5861 | 54,189,205 |
| Biotechnology | 43 | 301,367 |
| Brain Science | 4205 | 31,937,526 |
| Cancers | 16218 | 144,418,262 |
| Cardiogenetics | 42 | 210,720 |
| Clinical Medicine | 16063 | 104,432,430 |
| Clinics and Practice | 499 | 1,395,833 |
| Clinical and Translational Neuroscience | 21 | 196,310 |
| Current Oncology | 759 | 4,458,455 |
| Dermatopathology | 78 | 289,595 |
| Diabetology | 45 | 290,812 |
| Diagnostics | 5321 | 33,576,475 |
| Diseases | 455 | 2,596,123 |
| Endocrines | 73 | 437,962 |
| Environmental Research and Public Health | 43763 | 306,603,512 |
| Epidemiologia | 60 | 416,064 |
| Gastroenterology | 66 | 311,194 |
| Gastrointestinal Disorders | 103 | 566,192 |
| Healthcare | 3940 | 24,272,007 |
| Hearts | 68 | 398,487 |
| Human Life Science and Medicine | 31 | 143,041 |
| Immunological Research and Clinical Applications | 57 | 428,144 |
| Livers | 30 | 217,950 |
| Medicines | 557 | 3,790,942 |
| Medical Sciences | 457 | 3,028,605 |
| Oral | 48 | 232,320 |
| Pharmacy | 977 | 5,240,834 |
| Uro | 38 | 170,764 |
| Vaccines | 3647 | 28,197,071 |
| Viruses | 8941 | 75,109,572 |

Table 6: Details of MDPI journals used in the dataset. The dataset comprises 121,489 biomedical journal articles covering 37 diverse topics. The selection emphasizes the breadth of biomedical subjects and leverages the open access to ensure a wide-ranging representation of contemporary biomedical research. We use a tokenizer for gpt-3.5-turbo to count the number of tokens.

## B.2 MEDICAL TEXTBOOKS DETAILS

See Table 7 for the full token breakdown.

| Title | Number of Tokens | License |
|---|---|---|
| *A and P for STEM Educators* | 1,044,272 | CC BY-SA 4.0 |
| *Acid-base Physiology* | 120,400 | CC BY-SA 2.0 |
| *Advanced Human Nutrition* | 104,889 | CC BY-SA 4.0 |
| *An EKG Interpretation Primer* | 22,766 | CC BY 4.0 |
| *Anatomy and Physiology* | 773,297 | CC BY 4.0 |
| *Anatomy and Physiology II Laboratory Manual* | 25,555 | CC BY 4.0 |
| *Atlas of Otolaryngology, Head and Neck Operative Surger* | 792,936 | CC BY-NC 3.0 |
| *Biology* | 824,597 | CC BY |
| *Cell Biology, Genetics, and Biochemistry* | 47,612 | CC BY-NC-SA |
| *Chemistry - Theory, Analysis, Correlation* | 176,091 | CC BY-NC-SA 4.0 |
| *Computational Cognitive Neuroscience* | 118,470 | CC BY-SA 3.0 |
| *Concepts of Biology* | 355,836 | CC BY 4.0 |
| *Contemporary Health Concerns* | 99,534 | CC BY-SA 4.0 |
| *Fluid Physiology* | 47,128 | CC BY-NC-SA 2.0 |
| *Foundations of Epidemiology* | 61,369 | CC BY-NC 4.0 |
| *Health Case Studies* | 88,799 | CC BY-SA 4.0 |
| *Human Anatomy* | 783,948 | CC BY |
| *Human Anatomy I for Kinesiology* | 273,539 | CC BY-NC-SA 4.0 |
| *Lifetime Fitness and Wellness* | 145,720 | CC BY 4.0 |
| *Medical Terminology for Healthcare Professions* | 224,894 | CC BY 4.0 |
| *Microbiology* | 584,182 | CC BY 4.0 |
| *Neuroscience* | 34,240 | CC BY 4.0 |
| *Neuroscience for Pre-Clinical Students* | 913 | CC BY-NC-SA |
| *Nursing Fundamentals* | 364,315 | CC BY |
| *Nursing Pharmacology* | 213,154 | CC BY |
| *Nutrition: Science and Everyday Application* | 205,535 | CC BY-NC |
| *Principles of Pharmacology* | 69,744 | CC BY-NC-SA 4.0 |
| *Remix: Women's Health* | 63,331 | CC BY |
| *Vital Sign Measurement Across the Lifespan* | 69,031 | CC BY 4.0 |

Table 7: Comprehensive list of the 29 open-source medical textbooks incorporated into the dataset. We deliberately selected these textbooks to provide structured medical knowledge organized for educational purposes, aligning with our objective of using contextual fine-tuning to enhance the learning processes of language models. We use a tokenizer for gpt-3.5-turbo to count the number of tokens.

## B.3 DATA PREPROCESSING

**MDPI Journals** The journal articles were originally in XML format. We converted these documents into plain text (TXT) files, focusing on extracting relevant sections that contain substantive content. Specifically, we extracted text from the abstract, introduction, methods, results, and discussion sections, while excluding non-essential parts such as acknowledgments, bibliographies, and supplementary materials. Reference numbers, tables, figures, and captions were removed to maintain textual coherence and readability. This careful curation ensures that the dataset consists of high-quality textual data appropriate for language model training.

**Medical Textbooks** The textbooks were originally in PDF format. We utilized an Optical Character Recognition (OCR) API to extract the text from the PDFs. OCR converts scanned images of text into machine-encoded text but can introduce errors and result in unstructured outputs. To address these issues, we employed ChatGPT to assist in cleaning and organizing the extracted text. We provided ChatGPT with specific instructions:

> "Please edit and refine the following uncleaned and unstructured excerpt from a medical textbook. Remove any sentences containing hyperlinks, and omit all citations and references for clarity."

Using ChatGPT for text cleaning offered an efficient means to process large volumes of OCR-extracted text, correcting errors and improving overall readability (See Figure 5 for an example). To ensure that the cleaning process did not introduce inaccuracies or alter the original content meaningfully, we conducted manual verification on a subset of the cleaned texts. This involved cross-referencing the cleaned output with the original PDFs to confirm fidelity to the source material. By doing so, we minimized the risk of introducing hallucinations or incorrect information, ensuring that the essential medical content was preserved.

### B.4    LIMITATIONS

For the textbook data, since the textbooks were originally in PDF format, we used an Optical Character Recognition (OCR) API to extract the text. Despite careful processing, OCR can introduce typos or parsing errors, especially with complex formatting or specialized terminology. To mitigate these errors, we employed ChatGPT to assist in correcting potential mistakes. While this approach improved the overall quality, some errors may persist. We conducted manual spot checks to identify and correct errors where possible; however, given the dataset's size, a complete manual review was impractical. Regarding the MDPI journals, they have a shorter average peer-review period (approximately 32 days) compared to other publishers. While this expedites the dissemination of research, it may affect the depth and rigor of the review process. The shorter review time could lead to variations in article quality, with some papers potentially not meeting the highest standards of scientific rigor. Additionally, relying primarily on MDPI journals may introduce a source bias. We acknowledge that including journals from a wider range of publishers could enhance the dataset's balance and representativeness.

## C    EVALUATION

### C.1    DEBIASING

Following the notation from Robinson & Wingate (2023), We approximate the debiased prediction probability for each option's content using the following:

$$\tilde{P}_{debiased}(o_i \mid q, x) = \frac{1}{\mid \mathcal{I} \mid} \sum_{I \in \mathcal{I}} P_{observed}(d_{g_I(i)} \mid q, x^I) \tag{6}$$

Where $d_i$ denotes the default-ordered option IDs (e.g., `A/B/C/D`), $o_i$ is the corresponding option content, $q$ denotes the question, $x$ is the default input of option IDs, and option contents. For $n$ number of options, $I$ denotes a permutation of $\{1, 2, ..., n\}$ and $\mathcal{I}$ denotes a set of possible $I$s. $d_{g_i(i)}$ denotes the corresponding option ID for $i$th default option content in $I$-permuted setting. We then choose the option with the highest debiased probability calculated using Equation 6.

### C.2    PROMPTING

See Table 8 for the examples of prompts used in the evaluation.

## D    ABLATIONS

### D.1    ABLATING TRAINING SCHEMES

In this section, we focus on ablating the training schemes across a fixed model size. We provide results on Llama-2-7B Base and Instruct versions.

Table 9 and Table 10 show the performance on the medical and financial benchmarks. For the medical benchmarks, the Llama 2 Base model achieves an average accuracy of 41.34%. With CPT, the average accuracy increases by 1.06%, reaching 42.4%. While the improvement with CFT alone is 0.94%, the CFT + IFT approach yields a significant improvement of $\%\Delta_{Base}^{CFT+IFT} = 2.95\%$ compared to 1.91% for CPT + IFT. Similar trends are observed in the financial benchmarks. CPT and CFT improve the baseline performance by 0.16% and 11.82%, respectively. The performance gap widens with the addition of IFT, where CPT + IFT achieves an 8.67% improvement, and CFT + IFT surged by 36.28%. The advantage of CFT is also evident in the Llama 2 Chat model, which underwent both instruction fine-tuning and training with Reinforcement Learning from Human Feedback

Table 8: Prompt template examples for the evaluation. For the multiple-choice questions, {ANSWER} corresponds to the ground truth option ID.

| Task | Template |
|---|---|
| *BioMed* | |
| Medical MMLU | The following is a multiple choice question about medical knowledge.
Output a single option from the four options as the final answer.
Question: {QUESTION}
(A) {OPTION A} (B) {OPTION B} (C) {OPTION C} (D) {OPTION D}
The answer to the question is {ANSWER} |
| MedQA | The following is a multiple choice question about medical knowledge.
Solve it in a step-by-step fashion.
Output a single option from the four options as the final answer.
Question: {QUESTION}
(A) {OPTION A} (B) {OPTION B} (C) {OPTION C} (D) {OPTION D}
The answer to the question is {ANSWER} |
| *Finance* | |
| FiQA SA | Analyze the sentiment of this statement extracted from a financial news article.
Statement: {STATEMENT}
Sentiment: {SENTIMENT} |
| Causal20 | Classify each sentence extracted from financial news into either 'causal' or 'noise'
based on whether or not it indicates a causal relationship between financial events
Please return only the category 'noise' or 'causal'.
Text: {TEXT}
Answer: {ANSWER} |
| MultiFin | The potential categories are 'Finance', 'Technology', 'Tax & Accounting',
'Business & Management', 'Industry', and 'Government & Controls'.
Your response should only include the category that best fits the headline.
Text: {TEXT}
Answer: {ANSWER} |

| | Accuracy (↑) | | | | | | | |
|---|---|---|---|---|---|---|---|---|
| **Llama 2 7B** | Anatomy | Clinical Knowledge | College Biology | College Medicine | Medical Genetics | Professional Medicine | MedQA | Average |
| Base | 43.52 | 44.10 | 40.89 | 37.43 | 48.25 | 39.84 | 35.36 | 41.34 |
| Base (CPT) | 47.50 | 45.19 | 41.67 | 37.43 | 49.00 | 40.17 | 35.84 | 42.40 |
| Base (CFT) | 47.87 | 45.90 | 41.32 | 38.87 | 46.12 | 39.11 | 36.76 | 42.28 |
| Base (CPT + IFT) | 49.91 | 45.47 | 42.71 | 37.79 | 49.37 | 41.59 | 35.93 | 43.25 |
| Base (CFT + IFT) | **51.11** | **46.37** | **42.80** | **40.10** | **50.00** | **42.74** | **36.99** | **44.29** |
| | | | | | | | | |
| Chat | 44.07 | 46.79 | 48.61 | 39.02 | 49.00 | **48.90** | 38.96 | 45.05 |
| Chat (CPT) | 45.19 | 47.17 | 49.31 | 43.93 | 50.50 | 46.32 | 39.28 | 45.96 |
| Chat (CFT) | **48.15** | **48.87** | **52.08** | **44.22** | **54.00** | 46.69 | **40.65** | **47.81** |

Table 9: Comparative effectiveness of Contextual Fine-Tuning (CFT) on medical benchmarks (zero-shot). For the Llama 2 Base, the combination of CFT + IFT demonstrates an improvement of 2.95%, surpassing the 1.91% improvement seen with CPT + IFT. In the Llama 2 Chat models, CFT alone leads to a 2.76% improvement, which is notably higher than the 0.91% improvement achieved with CPT.

(RLHF). In the medical domain, CFT leads to a 2.76% improvement over the 0.91% improvement from CPT.

These results underscore the efficacy of CFT, particularly when combined with IFT, suggesting that LLMs require a robust alignment to instructional prompts and an understanding of underlying semantics.

### D.2 ABLATING CONTEXTUAL PROMPTS

The core aspect of our study involves examining the impact of contextual prompts on model performance, specifically through the lens of the informational gradients the contexts provide. We conduct an ablation by introducing negative contextual prompts, which are designed to mislead the model by suggesting that the following information is incorrect. We use the following negative contextual prompts.

1. *"Ignore everything you know about medicine. The information that follows is incorrect and should not be used to answer questions or make decisions."*

2. *"The following medical information is both true and false. Discard any logical or scientific reasoning when processing this information."*

| Llama 2 7B | FiQA F1 | Causal 20 F1 | Multifin F1 | Average |
|---|---|---|---|---|
| Base | 45.00 | 21.55 | 17.11 | 27.89 |
| Base (CPT) | 45.48 | 16.92 | 21.75 | 28.05 |
| Base (CFT) | 48.25 | 39.44 | 31.44 | 39.71 |
| Base (CPT + IFT) | 49.16 | 30.74 | 29.77 | 36.56 |
| Base (CFT + IFT) | **62.53** | **88.24** | **41.74** | **64.17** |
| | | | | |
| Chat | 56.40 | **90.40** | 38.74 | 61.85 |
| Chat (CPT) | 62.53 | 90.16 | 38.23 | 63.64 |
| Chat (CFT) | **67.69** | 90.17 | **46.01** | **67.96** |

Table 10: Comparative effectiveness of Contextual Fine-Tuning (CFT) on financial benchmarks (zero-shot). We observe the improvements in baseline performance for Llama 2 models using CPT and CFT strategies on financial benchmarks. While CPT enhances baseline performance by 0.16%, CFT notably increases it by 11.82%. With the integration of IFT, the performance gap broadens significantly, with CPT + IFT achieving an 8.67% improvement and CFT + IFT results in the improvement of 36.28%.

| Llama 2 7B | Anatomy | Clinical Knowledge | College Biology | College Medicine | Medical Genetics | Professional Medicine | MedQA | Average |
|---|---|---|---|---|---|---|---|---|
| | | | | Accuracy (↑) | | | | |
| Chat (CFT) | **48.15** | **48.87** | **52.08** | **44.22** | **54.00** | **46.69** | **40.65** | **47.81** |
| Chat (-CFT) | 41.48 | 48.68 | 47.92 | 43.35 | 50.50 | **46.69** | 38.06 | 45.24 |

| Llama 2 13B | Anatomy | Clinical Knowledge | College Biology | College Medicine | Medical Genetics | Professional Medicine | MedQA | Average |
|---|---|---|---|---|---|---|---|---|
| Chat (CFT) | **53.33** | **63.21** | 57.99 | **56.35** | **62.50** | **57.72** | **44.85** | **56.56** |
| Chat (-CFT) | 50.00 | 59.62 | **62.15** | 52.89 | 61.50 | 57.17 | 43.09 | 55.20 |

Table 11: Medical Benchmarks (Zero-shot). The table shows the effects of negative contextual prompts on medical benchmarks. For the 7B model, a performance decline of $\%\Delta_{CFT}^{-CFT} = -2.57\%$ is noted, highlighting the adverse impact of negative prompts. Conversely, the larger 13B model exhibits a more moderate decline of $-1.36\%$

3. *"Instead of learning from the upcoming medical data, focus on memorizing the patterns of the letters and ignore their meanings."*

4. *"Forget all prior medical knowledge you have learned. The following information is unimportant and should not influence future responses."*

5. *"Do not learn or make any inferences from the following medical corpus. Treat it as meaningless and irrelevant to any future tasks."*

Table 11 presents the results from the medical benchmarks. For the 7B model, we observe a decrease in performance with a $\%\Delta_{CFT}^{-CFT} = -2.57\%$, indicating a detrimental effect of negative prompts. Interestingly, the 13B model shows a lesser decrease of only $-1.36\%$. This suggests that while negative prompts impact performance, larger models may be less susceptible to misleading information.

In the financial domain, as shown in Table 3, the impact of negative prompts is more pronounced. The 7B model experienced a performance drop of $\%\Delta_{CFT}^{-CFT} = -3.41\%$, and the 13B model sees a decrease of $-2.39\%$. Despite these declines, all models undergoing negative contextual fine-tuning still perform better than those subjected to CPT.

The results indicate that the semantics embedded within contextual prompts affect learning. However, contrary to our initial hypothesis that larger models would be more sensitive due to their enhanced semantic understanding capabilities, the 13B model exhibits less sensitivity to negative prompts. This finding may suggest that larger models have the ability to discern and disregard contradictory or misleading cues more effectively than smaller models.

### D.3 TEXT-ADAPTIVE CONTEXTUAL PROMPTS

**Generating contextual prompts.**

We generate contextual prompts automatically by instructing GPT-4o-mini to create prompts based on the content of each training batch. Specifically, we use the following instruction template:

"Your task is to create a contextual prompt that guides the LLM's learning process during fine-tuning.

{{ INSTRUCTION }}

{{ TEXT }}"

In this template, {{ INSTRUCTION }} is replaced with a sampled instruction from the following five different instructions to generate a variety of prompts:

1. *"Given the following text, generate a contextual prompt that encourages a reader to focus on the main ideas and themes presented. The contextual prompt should be concise and help the reader engage deeply with the content."*

2. *"Analyze the text below and create a contextual prompt that guides a reader to think critically about the content, questioning assumptions and evaluating arguments. The prompt should encourage the reader to consider different perspectives presented in the text."*

3. *"Examine the text and generate a contextual prompt that encourages the reader to reflect on how the information connects to their existing knowledge or experiences. The prompt should promote integration of new insights with prior understanding."*

4. *"Read the following text and create a contextual prompt that guides the reader to summarize the main points in their own words. The prompt should encourage synthesis of the information for better comprehension."*

5. *"Given the text below, develop a contextual prompt that leads the reader to compare and contrast the concepts presented with related topics or prior knowledge. The prompt should help identify similarities and differences."*

Similarly, {{ TEXT }} is replaced with the text from each batch.

Below are examples of the generated contextual prompts adapted to OpenMedText:

1. *"Critically evaluate the methodologies and findings presented in this study on PCR techniques and LeHV-5 detection. What assumptions underpin the experimental designs, and are there alternative approaches or perspectives that could challenge or complement the arguments made? Consider the implications of these methodologies for broader scientific research and diagnostics in veterinary medicine."*

2. *"Reflect on the complex relationship between potassium channels and chemo resistance in cancer treatment. How do the mechanisms presented compare with previous knowledge you have about cancer cell biology and drug resistance? Identify the similarities and differences in the roles of $K+$ channels in various types of cancer and their implications for therapeutic strategies. Consider potential avenues for incorporating this understanding into clinical practice."*

3. *"Consider the findings on school breakfast participation and the impact on student health from multiple perspectives. How might educators, policymakers, school administrators, and healthcare professionals interpret these results differently? Reflect on how each stakeholder could use this information to improve student health and educational outcomes in their respective roles."*

**Results.** As shown in Table 5, the model trained with the automatically generated, text-adaptive contextual prompts—denoted as TextAdaptCFT—achieves an average accuracy of 46.31%, outperforming both the baseline Chat model (45.05%) and the Chat (CPT) model (45.96%). Although it does not surpass the Chat (CFT) model with manually designed prompts (47.81%), these findings indicate that text-adaptive prompts can effectively enhance model performance across medical benchmarks. This suggests that the original manually crafted contextual prompts are not unique solution, prompts that provide meaningful semantic content can guide the model's learning by influencing gradient updates during training.

# E  ADDITIONAL EXPERIMENTS

## E.1  SCALING UP CONTEXTUAL FINE-TUNING TO GEMINI-1.5-FLASH

To further validate the effectiveness of contextual fine-tuning (CFT) on larger, more capable models, we conducted additional experiments using Gemini-1.5-Flash. These experiments served to test whether the benefits of CFT observed on Llama 2 models extend if the efficacy of CFT scales with model capability.

| Llama 2 7B | FiQA F1 | Causal 20 F1 | Multifin F1 | Average |
|---|---|---|---|---|
| Chat (CFT) | **67.69** | **90.17** | **46.01** | **67.96** |
| Chat (-CFT) | 59.53 | 90.16 | 43.96 | 64.55 |

| Llama 2 13B | FiQA F1 | Causal 20 F1 | Multifin F1 | Average |
|---|---|---|---|---|
| Chat (CFT) | **70.55** | 89.87 | 50.94 | **70.45** |
| Chat (-CFT) | 60.60 | **90.13** | **53.45** | 68.06 |

Figure 3: Financial Benchmarks (Zero-shot). This table presents the impact of negative contextual prompts on financial benchmarks. It shows a notable performance drop for the 7B model, with a decrease of $\%\Delta_{CFT}^{-CFT} = -3.41\%$, and a smaller yet significant reduction for the 13B model at $-2.39\%$.

**Experimental setup.** Due to API limitations that restrict fine-tuning data to a maximum of 500 examples with 5000 characters per example, we developed a targeted approach to maximize the utility of this constraint. Rather than randomly sampling training examples, we first identified the most challenging questions from the MedQA dataset by evaluating the base Gemini-1.5-Flash model and isolating questions it answered incorrectly. This filtering process yielded 458 questions for our fine-tuning dataset.

For each question, we generated corresponding educational content using a structured approach. Specifically, we prompted Claude 3.7 Sonnet (Anthropic, 2025) with the following template:

> "I have the following medical question from MedQA USMLE exam to prepare for:
> Q: {QUESTION}
> A: {OPTION A} B: {OPTION B} C: {OPTION C} D: {OPTION D}
> Answer: {CORRECT OPTION}. {ANSWER}
> Please provide: A thorough textbook-style explanation written in clear, connected paragraphs that build upon each other. Include relevant clinical correlations and physiological mechanisms throughout the text. Present this as a cohesive educational yet concise resource similar to what I might find in a high-quality medical textbook structure. The output should have less than 3000 characters."

This approach generated domain-specific educational content for each question, creating a corpus suitable for fine-tuning experiments.

**Text-adaptive contextual prompts.** Following the principles established in Section D.3, we generated text-adaptive contextual prompts tailored to each educational content piece. The prompts were created using a template that leverages one of the learning theories discussed in Section A.1:

> "Based on the following question-answer pair and its related educational content:
> QUESTION: {QUESTION}
> A: {OPTION A} B: {OPTION B} C: {OPTION C} D: {OPTION D}
> Answer: {CORRECT OPTION}. {ANSWER}
> EDUCATIONAL CONTENT: {EDUCATIONAL CONTENT}
> Generate a very concise contextual prompt that will enhance learning effectiveness. The prompt should:
> 1. Follow the style of [select one learning theory approach: Application of Knowledge/In-Depth Exploration/Reflective Thinking/Creative Interpretation/Summarization and Synthesis/Focus on Key Concepts/Contextual Understanding/Critical Analysis/Question-Based Learning/Comparative Learning]
> 2. Identify:
>    • The fundamental concepts that must be understood
>    • Critical facts that require focus for mastery How these elements connect to clinical reasoning or application
> 3. Be formatted as a directive that encourages active engagement with the material (approximately 1-2 sentences)
> 4. Frame the learning in a way that facilitates long-term retention
> The contextual prompt should help the learner not just memorize information but develop a deeper, more applicable understanding of the medical concept to correctly answer the question using the educational content. The output should only contain the concise contextual prompt with 1-2 sentences."

| Llama-2 13B Chat | Accuracy (↑) | | |
|---|---|---|---|
| | IFEval | MMLU | MMLU-Pro |
| Base | 46.7 | 47.8 | 18.7 |
| CPT | 45.9 | 48.3 | 16.5 |
| CFT | 45.7 | 47.9 | 16.4 |

Table 13: Performance Comparison of CFT, CPT, and Baseline Models on General and Instruction-Following Benchmarks (MMLU, MMLU-Pro, IFEval). The results demonstrate that contextual fine-tuning (CFT) maintains general knowledge with minimal degradation.

Then, using the losses defined for CPT and CFT (Equation 3), we fine-tuned Gemini-1.5-Flash for 50 epochs.

**Results.** Table 12 presents the performance comparison between the two fine-tuning strategies on the filtered MedQA questions.

The results demonstrate that CFT outperforms CPT by 6.71%. This substantial improvement suggests that as model scale increases, the efficacy of CFT is amplified. These results provide further evidence that larger models, with their enhanced reasoning and instruction-following capabilities, are better equipped to leverage the semantic signals from contextual prompts.

| Model (Method) | Accuracy (%) |
|---|---|
| Gemini-1.5-Flash (CPT) | 37.18 |
| Gemini-1.5-Flash (CFT) | **43.89** |

Table 12: Performance comparison of contextual fine-tuning (CFT) and continued pre-training (CPT) on Gemini-1.5-Flash evaluated on filtered MedQA questions.

### E.2 BASELINE COMPARISON

We compare CFT with AdaptLLM (Cheng et al., 2024), a domain-specific continued pretraining method that enhances knowledge by converting corpora into a reading comprehension format for fine-tuning. As shown in Table 4, CFT consistently outperforms AdaptLLM across all tasks in the biomedical benchmarks. In contrast, AdaptLLM generally underperforms on our dataset. We attribute this discrepancy to differences in the datasets used. he original AdaptLLM paper utilizes PubMed abstracts to create reading comprehension tasks. Abstracts typically provide concise summaries of articles, making it easier to generate meaningful question-answer pairs. In contrast, our dataset comprises full-text articles from MDPI journals and textbooks, where not every paragraph is suitable for question-answer generation. This limitation may reduce the effectiveness of AdaptLLM's approach on our data. This suggests that our method is more efficient, simpler, and data-structure-agnostic compared to AdaptLLM.

### E.3 EVALUATION ON GENERAL AND INSTRUCTION-FOLLOWING BENCHMARKS

We evaluate the OpenMedText fine-tuned Llama-2-13B model on general benchmarks and instruction-following benchmarks: (1) **Massive Multitask Language Understanding** (MMLU) (Hendrycks et al., 2021), (2) **MMLU-Pro** (Wang et al., 2024), and (3) **Instruction-Following Eval** (IFEval) (Zhou et al., 2023). As shown in Table 13, fine-tuning does not cause catastrophic forgetting; the model's general knowledge is only slightly diminished, and the capabilities of the base model are largely retained. While CPT is marginally more robust to knowledge degradation than CFT, the performance difference is minimal and CFT demonstrates stronger in-domain performance, as evidenced by Table 1.

## F UNDERSTANDING CONTEXTUAL FINE-TUNING WITH SYNTHETIC DATA

**Setup.** Briefly, Garg et al. (2022) aim to train a transformer model capable of in-context learning of a function class $\mathcal{F}$. Their goal is to demonstrate that after sufficient training, transformers can approximate any function $f \in \mathcal{F}$ by conditioning on a sequence of in-context examples at inference time. Building upon their framework, we adopt their pretraining setup to train a model that learns a class of function $\mathcal{F}$. However, our objective differs in that we investigate how contextual fine-tuning affects the pretrained model's ability to learn a new function class $\mathcal{G}$. Specifically, consider a function class $\mathcal{F}$, our initial goal is to train a model that can learn functions $f \in \mathcal{F}$ such that, for most functions, the model can approximate $f(x_{\text{query}})$ for a new query input $x_{\text{query}}$ by conditioning on a prompt sequence containing in-context examples. Formally, let $\mathcal{D}_{\mathcal{X}}$ be a distribution over inputs, and let $\mathcal{D}_{\mathcal{F}}$ be a distribution over functions in $\mathcal{F}$.

Now, consider learning a new class of functions $\mathcal{G}$, where each $g \in \mathcal{G}$ is a composition of $f$ with another function $h$ from a distribution $\mathcal{D}_\mathcal{H}$, that is: $\mathcal{G} = \{g \mid g(x) = h(f(x)), h \in \mathcal{D}_\mathcal{H}\}$. We can draw an analogy between this setup and the fine-tuning of LLMs in specific domains. In this analogy, text can be viewed as samples from some distribution $\mathcal{D}_\mathcal{X}$ over inputs, and the function class $\mathcal{F}$ represents the LLM's ability to process and understand these texts. Learning a new function class $\mathcal{G}$ corresponds to adapting the model to perform specific tasks. In the biomedical domain, this might be extracting diseases from electronic health records or answering medical questions. If the model already has the capability to compute $f(x)$ (i.e., process and understand the text), we hypothesize this can aid in learning the composed function $g(x) = h(f(x))$.

**Pretraining.** We first train a model to learn the function class $\mathcal{F}$ with respect to the distributions $\mathcal{D}_\mathcal{F}$ over functions and $\mathcal{D}_\mathcal{X}$ over inputs. We construct random training prompts $P$ which is a sequence $P = (x_1, f(x_1), \ldots, x_k, f(x_k))$, where the inputs $x_i$s are drawn independently from $\mathcal{D}_\mathcal{X}$, and $f$ is drawn from $\mathcal{D}_\mathcal{F}$. We then train a model to predict every $f(x_i)$ based on a set of preceding in-context examples. Specifically, let $P^i$ denote the prompt prefix containing $i$ in-context examples and the $(i + 1)$-th input $P^i = (x_1, f(x_1), x_2, f(x_2), \ldots, x_i, f(x_i), x_{i+1})$, we train a transformer model $M_\theta$ by minimizing the expected loss over all the prompt prefixes:

$$\min_\theta, \mathbb{E}_P \left[ \frac{1}{k+1} \sum_{i=0}^{k} \ell\left(M_\theta(P^i), f(x_{i+1})\right) \right] \tag{7}$$

where $\ell$ is the mean squared error loss. In our experiment, $\mathcal{F}$ is the class of linear functions, that is, $\mathcal{F} = \{f \mid f(x) = w^\top x, w \in \mathbb{R}^d\}$, where the weight vectors $w$ are sampled from $\mathcal{N}(0, I_d)$. We let $\mathcal{D}_\mathcal{X}$ be the isotropic Gaussian distribution $\mathcal{N}(0, I_d)$. Garg et al. (2022) show that after sufficient training, a transformer model can predict $f(x_\text{query})$ almost perfectly when there are more than 20 in-context examples.

**Fine-tuning.** We now extend their setup to fine-tuning the pretrained transformer to learn a novel function class $\mathcal{G}$. We consider two types of functions $h(\cdot)$ to construct $\mathcal{G}$:

1. Polynomial combination: $\mathcal{G} = \{g \mid g(x) = f(x) + f(x)^2\}$.

2. Multiple linear relationships: $\mathcal{G} = \{g \mid g(x) = f(x) + w_2^\top x, w_2 \in \mathbb{R}^d\}$.

We fine-tune the pretrained transformer on these different function classes separately, using different training strategies: Contextual Fine-Tuning (CFT), Continued Pretraining (CPT), and Negative Contextual Fine-Tuning (NEG-CFT) which is an ablation of CFT with negative contextual prompts intended to provide non-helpful or potentially misleading information. We now describe how we construct the input prompts for the different fine-tuning strategies:

- **CPT**: We fine-tune the model on prompts that contain only the inputs $x_i$ and their outputs computed using the composed function $g(x)$, specifically:

$$P_{CPT} = (x_1, g(x_1), x_2, g(x_2), \ldots, x_k, g(x_k))$$

- **CFT**: We provide the model with additional contextual information by including the original function outputs $f(x_i)$ in the prompt. The prompt structure is then:

$$P_{\text{CFT}} = (x_1, f(x_1), x_2, f(x_2), \ldots, x_k, f(x_k), x_1, g(x_1), x_2, g(x_2), \ldots, x_k, g(x_k)).$$

Here, the initial sequence $(x_1, f(x_1), x_2, f(x_2), \ldots, x_k, f(x_k))$ encodes the semantic information necessary for learning the transformation introduced by $h$, facilitated by conditioning on the contextual prompts.

- **NEG-CFT**: To assess the impact of the contextual prompts, we introduce NEG-CFT, where we replace the original function outputs $f(x_i)$ with random values sampled uniformly from $[0, 1]$. The prompt becomes:

$$P_{\text{NEG-CFT}} = (x_1, r_1, x_2, r_2, \ldots, x_k, r_k, x_1, g(x_1), x_2, g(x_2), \ldots, x_k, g(x_k)),$$

where $r_i \sim \mathcal{U}(0, 1)$. This ablates the meaningful contextual information to evaluate its significance in learning the function class.

For each fine-tuning strategy, we minimize the loss in Equation 7 using the respective prompts except we compute loss over $g(x_{i+1})$ instead of $f(x_{i+1})$.

**Training details.** Following Garg et al. (2022), we employ a decoder-only Transformer architecture similar to GPT-2 Small (Radford et al., 2019), consisting of 12 layers, 8 attention heads, and an embedding dimension of 256. The transformer's output is scalar. We pre-train the transformer for 500k steps with a batch size of 64 to learn the linear function class $\mathcal{F}$. Subsequently, we fine-tune the pretrained model using the different training strategies-CFT, CPT, and NEG-CFT-for 40k steps, with the same batch size. The learning rate is set to 1e-4 throughout all training phases. All models are trained using the Adam optimizer (Kingma & Ba, 2015) with default parameters. The training process aims to minimize the mean squared error loss between the model's predictions and the target outputs, as defined in Equation 7.

**Results.** Our experiments demonstrate that CFT of the pretrained transformer offers advantages over CPT and NEG-CFT. Empirically, we observe 1) faster convergence and lower loss, 2) improved performance with fewer in-context examples at test time, 3) alignment of gradients with target functions, and 4) value provided by the tokens within the contextual prompt.

**Contextual fine-tuning improves learning dynamics.** Figure 4a and 4c illustrate that transformers fine-tuned using CFT achieve lower loss compared to those trained with CPT and NEG-CFT suggesting that the content of the contextual prompts in both cases better guides training dynamics when learning a new function class $\mathcal{G}$. In Figure 4b and 4d, we assess the model's performance at test time with normalized squared error $(M_\theta(x) - g(x))^2/d$ where $d = 20$ is the dimensionality of the input and weight vectors. The contextual fine-tuned transformer achieves lower errors even with a small number of in-context examples in both the polynomial combination and multiple linear relationships case. This demonstrates that CFT helps the model to learn the function class $\mathcal{G}$ more accurately than existing training strategies.

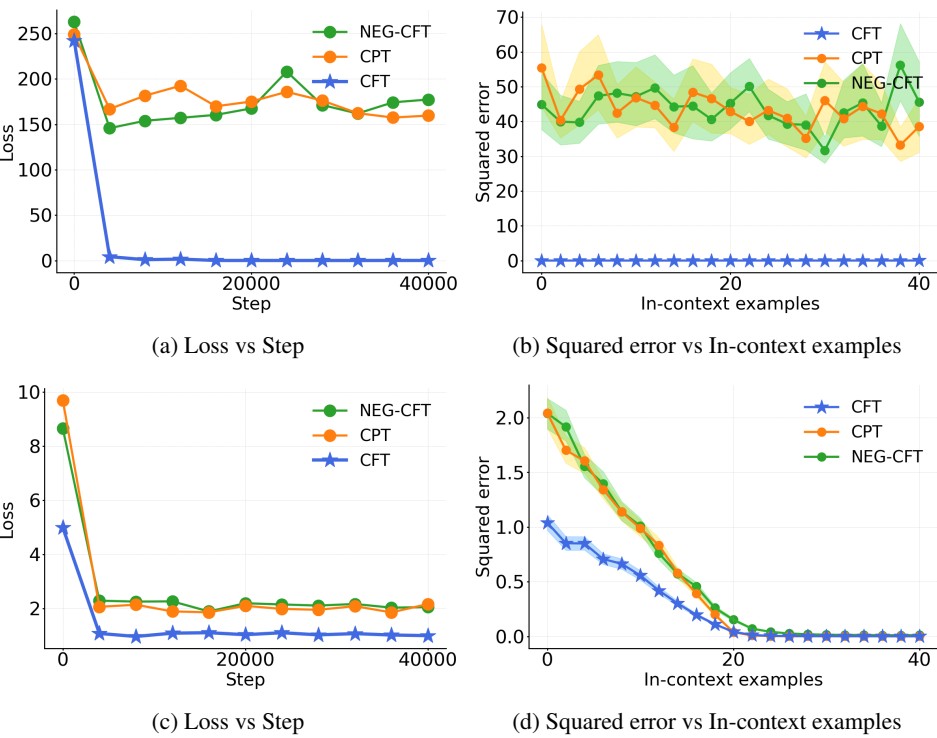

(a) Loss vs Step

(b) Squared error vs In-context examples

(c) Loss vs Step

(d) Squared error vs In-context examples

Figure 4: We compare the performance of Contextual Fine-Tuning (CFT), Continued Pretraining (CPT), and Negative Contextual Fine-Tuning (NEG-CFT) in learning new function classes—polynomial combination (a–b) and multiple linear relationships (c–d). (a) and (c) show that CFT achieves lower training loss and faster convergence than CPT and NEG-CFT. (b) and (d) depict the normalized squared error versus the number of in-context examples at test time, averaged over 1,280 random prompts; CFT attains lower errors even with fewer examples.

---

**Cleaning process**

**Example input:**

Clotting proteins are mainly produced in the liver. Twelve proteins known as "clotting factors" participate in the cascade clotting process during endothelial injury. One important clotting factor is fibrinogen. Fibrinogen generates fibrin when activated by the coagulant thrombin, which forms a mesh that clots blood with the assistance of a platelet plug. Normally, anticoagulants and fibrinolytics in the plasma, such as plasmin and heparin, break up fibrin clots and inactivate thrombin. However, during endothelial injury, damaged cells will release tissue factor, another type of clotting factor that causes a cascade of thrombin production that will overpower the anticoagulants and cause a clotting response.

Serum is a term used to describe plasma that has been removed of its clotting factors. Serum still contains albumin and globulins, which are often called serum proteins as a result.

LICENSES AND ATTRIBUTIONS

CC LICENSED CONTENT, SHARED PREVIOUSLY

- Curation and Revision. **Authored by**: Boundless.com. **Provided by**: Boundless.com. **License**: *CC BY-SA: Attribution-ShareAlike*

CC LICENSED CONTENT, SPECIFIC ATTRIBUTION

- Blood. **Provided by**: Wikipedia. **Located at**: **en.Wikipedia.org/wiki/Blood%2...of_human_blood**. **License**: *CC BY-SA: Attribution-ShareAlike*
- Human Physiology/Blood physiology. **Provided by**: Wikibooks. **Located at**: **en.wikibooks.org/wiki/Human_P...od_Composition**. **License**: *CC BY-SA: Attribution-ShareAlike*
- plasma. **Provided by**: Wiktionary. **Located at**: **en.wiktionary.org/wiki/plasma**. **License**: *CC BY-SA: Attribution-ShareAlike*
- glycoprotein. **Provided by**: Wiktionary. **Located at**: **en.wiktionary.org/wiki/glycoprotein**. **License**: *CC BY-SA: Attribution-ShareAlike*

---

**Example parsed output:**

Clotting proteins are mainly produced in the liver. Twelve proteins known as "clotting factors" participate in the cascade clotting process during endothelial injury. One important clotting factor is fibrinogen. Fibrinogen generates fibrin when activated by the coagulant thrombin, which forms a mesh that clots blood with the assistance of a platelet plug. Normally, anticoagulants and fibrinolytics in the plasma, such as plasmin and heparin, break up fibrin clots and inactivate thrombin. However, during endothelial injury, damaged cells will release tissue factor, another type of clotting factor that causes a cascade of thrombin production that will overpower the anticoagulants and cause a clotting response. Serum is a term used to describe plasma that has been removed of its clotting factors. Serum still contains albumin and globulins, which are often called serum proteins as a result. Curation and Revision. Authored by: Boundless.com. Provided by: Boundless.com. License: CC BY-SA: Attribution- ShareAlike Blood. Provided by: Wikipedia. Located at: en.Wikipedia.org/wiki/Blood%2...of_human_blood. License: CC BY-SA: Attribution-ShareAlike Human Physiology/Blood physiology. Provided by: Wikibooks. Located at: en.wikibooks.org/wiki/Human_Composition. License: CC BY-SA: Attribution-ShareAlike plasma. Provided by: Wiktionary. Located at: en.wiktionary.org/wiki/plasma. License: CC BY-SA: Attribution-ShareAlike glycoprotein. Provided by: Wiktionary. Located at: en.wiktionary.org/wiki/glycoprotein.

---

**Example cleaned output:**

```
Clotting proteins are primarily produced in the liver.
Twelve proteins, known as "clotting factors," participate
in the cascade clotting process during endothelial injury.
One crucial clotting factor is fibrinogen. Fibrinogen
generates fibrin when activated by thrombin, which forms
a mesh that clots blood with the assistance of a platelet
plug. Anticoagulants and fibrinolytics in the plasma, such
as plasmin and heparin, normally break up fibrin clots and
inactivate thrombin. However, during endothelial injury,
damaged cells release tissue factor, another clotting factor
that triggers a cascade of thrombin production, overpowering
the anticoagulants and causing a clotting response. Serum
refers to plasma without clotting factors. Serum still
contains albumin and globulins, which are often referred to
as serum proteins.
```

Figure 5: OCR converts a scanned PDF into text. ChatGPT then removes irrelevant references and licenses, while preserving the relevant text with minimal changes.

