# OpenReview forum: "Teaching LLMs How to Learn with Contextual Fine-Tuning"
_ICLR.cc/2025/Conference — ICLR 2025 Poster_

### Official Review · Reviewer_eqbM · 2024-11-02

**Soundness:** 3
**Presentation:** 3
**Contribution:** 3
**Rating:** 8
**Confidence:** 4

**Summary:**

The paper introduces a new technique, "contextual fine-tuning," which is designed to enhance fine-tuning of LLMs. The core idea is to prepend special text suggesting that the reader engage in deeper ways with upcoming content. The authors provide experimental evidence meant to show that the idea works in both synthetic and realistic settings.

**Strengths:**

The paper presents a very simple and intriguing idea for enhancing fine-tuning: prepending a prompt asking the reader to engage deeply with upcoming content.Moreover, experimental evidence from two real-world domains indicates that this technique outperforms natural baselines in several cases.

At a practical level, the experimental results are quite promising. This technique would be extremely easy to implement, so even small improvements with it would be a win. Although the experimental data indicates that the new method  ("Contextual Fine-Tuning" or CFT) is not always a win over the baseline over simply continued pretraining, it does seem to produce better results in many cases. Overall, I appreciated the medical and financial domain experiments, which seemed to have an appropriate level of complexity and realism to be a good test of the method.

At a theoretical level, this is certainly interesting. I find it counterintuitive that this technique works at all: one would think that the best way for the system to do next-token prediction would be simply to ignore the prepended CFT prompt, since it has no direct connection with the upcoming text. The fact that CFT produces good results is a hint that something surprising is happening under the hood, which could lead to future theoretical advances. Moreover, establishing that prepending this type of prompt is useful seems to point to many future areas of investigation for prepending other types of prompts.

UPDATE: In light of the new experiments, I have raised my score.

**Weaknesses:**

There is one major weakness in the paper: I am not at all convinced by Section 4. However, I think the best thing is for the authors simply to cut this section, because I actually don't think it's necessary for their argument.

Here's why I don't think Section 4 is relevant. In the synthetic function setting, the prepended prompts contain customized relevant information for the subsequent prediction tasks, helpfully factoring the functions that the system is learning to compute. This contrasts with the real-world CFT case, where—if I understand correctly—the prepended prompt has no customized connection at all to the upcoming text. I am happy to be corrected if I'm mistaken on this difference, but if I'm reading correctly, I think the synthetic task just doesn't shed light on the real-world task. That said, I don't think the synthetic task is particularly necessary for the argument, so I recommend simply cutting this section completely.

In addition to this major weakness, there are some aspects of exposition I'd try to improve. The first paragraph of the intro seems unnecessarily "hype-y." I actually think you could cut this paragraph and the paper would be fine. Also in the abstract and intro, I would say it's not clear to me that this method is necessarily "learning to learn": that might be one explanation, but there may be others. In the conclusion, the two paragraphs seem to repeat each other a bit. The authors might consider cutting the first one and just keeping the second. I also think it might be interesting for the authors, at some point, to speculate a bit on the mechanisms that make the technique work, or suggest some future experiments that might elucidate these mechanisms.

Typos: "human's" in abstract; "differes" and "initiial" around line 245.

**Questions:**

My main question is whether my reading of the difference between the synthetic vs. the real-world setting is correct. If the real-world prompts contain customized information about the upcoming text (contrary to my reading), then I would say Section 4 is relevant after all.

---

> ### Author Response · Authors · 2024-11-23
> **Response to Weakness: Synthetic Experiment and Question 1**
>
> **Weakness 1: Synthetic experiment and Question 1**
> >Here's why I don't think Section 4 is relevant. In the synthetic function setting, the prepended prompts contain customized relevant information for the subsequent prediction tasks, helpfully factoring the functions that the system is learning to compute. This contrasts with the real-world CFT case, where—if I understand correctly—the prepended prompt has no customized connection at all to the upcoming text. I am happy to be corrected if I'm mistaken on this difference, but if I'm reading correctly, I think the synthetic task just doesn't shed light on the real-world task. That said, I don't think the synthetic task is particularly necessary for the argument, so I recommend simply cutting this section completely.
>
> >My main question is whether my reading of the difference between the synthetic vs. the real-world setting is correct. If the real-world prompts contain customized information about the upcoming text (contrary to my reading), then I would say Section 4 is relevant after all.
>
> Thank you for your thoughtful comment. First, you are correct that, in the real-world setting, the prepended contextual prompts do not have a customized connection to the upcoming text. However, we'll highlight that we have newer experiments in Table 1 in the general comments that continue to indicate improved performance against CPT. Both original contextual fine-tuning and fine-tuning with contextual prompts that are dependent on the upcoming text (AutoDep-CFT) outperform the base Chat model and CPT across most tasks. i.e. even in experimental settings where real-world prompts contain customized information, CFT continues to outperform CPT. We hope the additional experiment, that we will fold into the manuscript, alleviates concern about the potential disconnect with respect to Section 4.
>
> That said, your comment also serves as a reminder to better motivate our synthetic experiment and our response to the reviewer TQCm to their question might be helpful.
>
> Our current hypothesis for why our approach works is that gradients under prompts that contain semantic content relevant for learning serve to regularize the process of learning via fine-tuning. However testing this hypothesis directly is challenging since (a) different LLMs might interpret semantic information in a prompt differently (as a function of scale) and (b) it requires knowing which neurons are responsible for representing the inferred semantic information in the prompt -- an open problem in mechanistic interpretability.
>
> The primary objective of the synthetic experiment was to analyze how contextual prompts affect the gradients of transformer models during training in a controlled setting where we can describe the semantic information that is necessary for learning explicitly via text. The advantage of this is that it enables us to not worry about how the transformer encodes semantic information (thus enabling the study of this phenomenon on much smaller models) and consequently better understand what properties of the gradient enable this.
>
> To expand on this further, the sequence of tokens we use in the synthetic data, by design, $(x_1,f(x_1),x_2,f(x_2),\ldots,x_k,f(x_k))$ encode the semantic information necessary for learning this synthetic class of problem, facilitated by conditioning on the prompts.

---

> ### Author Response · Authors · 2024-11-23
> **Response to Weakness: Writing**
>
> **Weakness 2: Writing**
> >In addition to this major weakness, there are some aspects of exposition I'd try to improve. The first paragraph of the intro seems unnecessarily "hype-y." I actually think you could cut this paragraph and the paper would be fine. Also in the abstract and intro, I would say it's not clear to me that this method is necessarily "learning to learn": that might be one explanation, but there may be others. In the conclusion, the two paragraphs seem to repeat each other a bit. The authors might consider cutting the first one and just keeping the second.
>
> Thank you sincerely for carefully reading our manuscript and for providing such thoughtful and constructive feedback.
>
> We will correct the typos, revise the abstract, the introduction and the conclusion as you suggest.
>
> > I also think it might be interesting for the authors, at some point, to speculate a bit on the mechanisms that make the technique work, or suggest some future experiments that might elucidate these mechanisms.
>
> In our response to the reviewer ULmj, we referenced the work "Why Think Step by Step? Reasoning Emerges from the Locality of Experience" by Prystawski et al. (2023). This study suggests that chain-of-thought (CoT) prompting works because local reasoning steps are embedded in the pretraining data, effectively simulating an internal thought process during training that manifests at inference time.
>
> Drawing inspiration from this, we hypothesize that our Contextual Fine-Tuning (CFT) method may work by leveraging similar mechanisms. The contextual prompts could be guiding the model to process information more effectively during training, influencing its internal representations and gradient updates.
>
> We believe that a future study examining how CFT influences the model's learning dynamics—as a function of the pretraining data—could shed light on the mechanisms behind the effectiveness of contextual prompts.
>
> Thank you again for your insightful suggestion.
>
> References:
>
> [1] Ben Prystawski and Noah D. Goodman. 2023. Why think step-by-step? Reasoning emerges from the locality of experience. CoRR abs/2304.03843 (2023)

---

### Official Review · Reviewer_TQCm · 2024-11-03

**Soundness:** 3
**Presentation:** 3
**Contribution:** 3
**Rating:** 8
**Confidence:** 3

**Summary:**

This paper introduces Contextual Fine-Tuning (CFT) for enhancing LLMs' learning capabilities by incorporating educational cognitive strategies during training. The key innovation is using instructional prompts designed to mimic human learning approaches, guiding the model's semantic understanding and domain-specific knowledge acquisition. The authors demonstrate CFT's effectiveness through experiments in medical and financial domains, showing improved performance compared to standard fine-tuning approaches while requiring limited training data.

**Strengths:**

- A novel contextual fine-tuning framework that combines in-context learning with gradient-based learning using educational psychology-inspired prompts
- Theoretical and empirical analysis demonstrating how contextual prompts affect model learning using synthetic experiments with simplified models
- Creation and curation of OpenMedText dataset, combining academic medical journals and textbooks, offering diverse training material
- Improved performance across down-stream applications, including medical and financial domains

**Weaknesses:**

- For the experimental settings, the main experiments focus on comparison between CFT and CPT. To demonstrate the effectiveness, shall the comparisons also include other ICL methods, also RAG-based methods?
- While the authors provide thoughtful prompts based on educational theories in Appendix B1, it seems to be very limited exploration of prompt optimization or automated prompt generation methods, as the prompt template seems very various and task-specific.
- It might be beneficial to include computational cost etc. for efficiency evaluation.
- While OpenMedText is a comprehensive dataset proposed in a research paper (not dataset/benchmark papers), more information regarding statistics, potential biases, quality issues etc. in the dataset are not thoroughly discussed.

**Questions:**

See above.

---

> ### Author Response · Authors · 2024-11-23
> **Response to Weakness 1**
>
> Thank you for your valuable feedback. We are happy to address your comments.
>
> **Weakness 1: Baseline comparison**
> >For the experimental settings, the main experiments focus on comparison between CFT and CPT. To demonstrate the effectiveness, shall the comparisons also include other ICL methods, also RAG-based methods?
>
> We agree that including evaluation of other In-Context Learning (ICL) methods and Retrieval-Augmented Generation (RAG)-based models would provide additional insights. However, ICL is only viable for models with very large context length and RAG based methods keep track of the entire content at inference time. These methodological differences make apples-apples direct comparisons challenging. Furthermore, both of these methods could also be used on a model that has been fine-tuned with CFT and so we see these methods as complementary and are looking to explore their use with CFT trained models in future work.
>
> To expand our set of baselines however, we have included AdaptLLM [2] as an additional baseline to provide a more comprehensive evaluation. AdaptLLM is better aligned methodologically with our approach, as it enhances domain-specific knowledge during training by converting specialized corpora into a reading comprehension format for fine-tuning. We evaluated our method against AdaptLLM on several medical benchmarks. The results are presented in the table below:
>
> **Table. Comparison against AdaptLLM**
> | Llama-2-7B | Anatomy   | Clinical Knowledge | College Biology | College Medicine | Medical Genetics | Professional Medicine | MedQA 	| Average   |
> |------------|-----------|--------------------|-----------------|------------------|------------------|-----------------------|-----------|-----------|
> | Chat   	| 44.07 	| 46.79          	| 48.61       	| 39.02        	| 49.00        	| **48.90**         	| 38.96 	| 45.05 	|
> | Chat (CPT) | 45.19 	| 47.17          	| 49.31       	| 43.93        	| 50.50        	| 46.32             	| 39.28 	| 45.96 	|
> | Chat (CFT) | **48.15** | **48.87**      	| **52.08**   	| **44.22**    	| **54.00**    	| 46.69             	| **40.65** | **47.81** |
> | AdaptLLM   | 44.45 	| 47.36          	| 48.27       	| 39.60        	| 45.00        	| 38.61             	| 37.12 	| 42.92 	|
>
> We conclude that:
> 1. Our CFT method consistently outperforms AdaptLLM across all tasks.
> 2. AdaptLLM did not perform as well as anticipated on our dataset. We speculate that this may be due to differences in the datasets used. The original AdaptLLM paper utilizes PubMed abstracts to create reading comprehension tasks. Abstracts typically provide concise summaries of articles, making it easier to generate meaningful question-answer pairs. In contrast, our dataset consists of full-text articles from MDPI journals and textbooks, where not every paragraph contains information that readily lends itself to question-answer generation. This may limit the effectiveness of AdaptLLM's methodology when applied to our dataset.
> 3. As AdaptLLM is the most recent work closely related to our approach, these results suggest that our CFT method provides superior performance in domain-specific fine-tuning.
>
> We hope that this additional comparison provides further confidence in the value of our method relative to existing work.

---

> ### Author Response · Authors · 2024-11-23
> **Response to Weakness 2**
>
> **Weakness 2: Limited exploration of prompt generation**
> >While the authors provide thoughtful prompts based on educational theories in Appendix B1, it seems to be very limited exploration of prompt optimization or automated prompt generation methods, as the prompt template seems very various and task-specific.
>
> We appreciate your recognition of the thoughtful prompts based on educational theories presented in Appendix B1 and we understand your concern about the limited exploration of prompt optimization and the task-specific nature of our prompt templates.
>
> Our primary objective in this paper was to explore the capability of incorporating contextual prompts during the training phase of language models, specifically through Contextual Fine-Tuning (CFT) rather than identifying the optimal set of prompts that, with CFT, would yield the highest improvements.
>
> Posing the identification of optimal prompts as an optimization problem is difficult. Prompts influence the entire learning trajectory of the model, affecting model weights and internal representations over many training rendering gradient-based and few-shot learning based methods for prompt optimization computationally infeasible.
>
> Please also see Table 1 in the general comments. We attempted an initial exploration of automated prompt generation to address your concerns. The results demonstrate that contextual fine-tuning with automatically generated prompts outperforms continued pre-training and is on par with our original contextual fine-tuning using hand-crafted prompts. This suggests that automatic prompt generation, even without explicit optimization, can produce effective contextual prompts. It also indicates that the content of the contextual prompts does not need to be unique or manually designed to enhance the model's performance.
>
> Our findings indicate that such methods can be a viable alternative to manually crafting prompts, potentially simplifying the fine-tuning process and making it more accessible. We believe this contributes to the broader understanding of how automated approaches can be employed in prompt design and optimization. We will include these new results and discussions in the revised version of our paper to provide a more comprehensive exploration of prompt generation methods. We are grateful for your feedback, which has helped us enhance our work and consider new avenues for research.

---

> ### Author Response · Authors · 2024-11-23
> **Response to Weakness 3 and 4**
>
> **Weakness 3: Including computational cost**
> >It might be beneficial to include computational cost etc. for efficiency evaluation.
>
> We have added the following paragraph in Appendix D.2 in blue, which will be included in the revised version of the paper.
>
> 	To assess the efficiency of CFT, we carefully measured the computational resources required for our experiments and compared the overhead introduced by incorporating contextual prompts. Below are the details of our computational setup and findings. We utilized the Fully Sharded Data Parallel (FSDP) training to efficiently distribute the model across multiple GPUs. Training was performed using the bf16 (Brain Floating Point) data format. We implemented Flash Attention 2. All training was conducted with 8 NVIDIA A100 GPUs. With the above configuration, we achieved a training speed of approximately 55,188 tokens per second, measured using the Llama tokenizer. The fine-tuning required a total of approximately 111.11 GPU-hours to complete. Incorporating contextual prompts increased the total training time by approximately 0.89 GPU-hours, resulting in a total of 112 GPU-hours. Each contextual prompt added only about 0.8% to the length of each training example on average. This slight increase in input length led to less than a 1% increase in total training time.
>
> **Weakness 4: More information about OpenMedText**
> >While OpenMedText is a comprehensive dataset proposed in a research paper (not dataset/benchmark papers), more information regarding statistics, potential biases, quality issues etc. in the dataset are not thoroughly discussed.
>
> In addition to the total number of tokens, we added a detailed breakdown of the number of journals in each category.
>
> We acknowledge that OpenMedText may have inherent limitations and potential biases, which we have added the following paragraph in Appendix C.4, which will be included in the revised version of the paper.
>
> 	For the textbook data, since the textbooks were originally in PDF format, we used an Optical Character Recognition (OCR) API to extract the text. Despite careful processing, OCR can introduce typos or parsing errors, especially with complex formatting or specialized terminology. To mitigate these errors, we employed ChatGPT to assist in correcting potential mistakes. While this approach improved the overall quality, some errors may persist. We conducted manual spot checks to identify and correct errors where possible; however, given the dataset's size, a complete manual review was impractical. Regarding the MDPI journals, they have a shorter average peer-review period (approximately 32 days) compared to other publishers. While this expedites the dissemination of research, it may affect the depth and rigor of the review process. The shorter review time could lead to variations in article quality, with some papers potentially not meeting the highest standards of scientific rigor. Additionally, relying primarily on MDPI journals may introduce a source bias. We acknowledge that including journals from a wider range of publishers could enhance the dataset's balance and representativeness.
>
> Thank you very much for your suggestion to provide more details.
>
> References:
>
> [2] Daixuan Cheng, Shaohan Huang, and Furu Wei. Adapting large language models via reading comprehension. arXiv:2309.09530, 2023.

---

> > ### Comment · Reviewer_TQCm · 2024-11-23
> >
> > Thank you for your reply. Most of my concerns have been solved and I will increase my score.

---

### Official Review · Reviewer_ULmj · 2024-11-03

**Soundness:** 2
**Presentation:** 3
**Contribution:** 3
**Rating:** 6
**Confidence:** 4

**Summary:**

This paper examines the role of prompting in effectively fine-tuning LLMs for new domains. Based on the premise that prompting can play a decisive role at inference time, this paper proposes the contextual fine-tuning approach by which an additional context prompt prefix is added before fine-tuning/continual pertaining documents. These contextual prompt prefixes are designed based off of human educational theories and the present 5 such examples in their work. In a synthetic function approximation setting, they perform some illustrative experiments demonstrating that the choice of fine-tuning prompt can significantly impact whether new functions can be added in post-pretraining. In this setting, they also introduce an ablative method, negative contextual fine-tuning in which the model is given random information in the context prefix and they show that this performs wors. They then examine a real domain adaptive setting, involving fine-tuning an LLM on medical and financial data. They demonstrate that their method performs better than existing approaches such as continual pertaining, instruction fine-tuning, and their combination. They also show that a "negative contextual prompt" which suggests that the provided information might be incorrect performs worse.

**Strengths:**

Overall this paper does examine an interesting and important problem of expanding the knowledge or capabilities of a pre-trained large language model on new domains or topics. They propose a surprising source of gains: simply adding a prompt prefix to domain-relevant documents can improve the data efficiency of learning these new domains. The experiments across two domains appear to be well-designed and show some gains. As they note in their paper, the additional annotation and computational burden of this method is seemingly non-existent, and it could be flexibly combined with many existing continual pre-training corpora and techniques. The also attempt to justify the sources of their gains using a synthetic function-approximation task and demonstrate how an additional context can  result in faster learning. Overall, this analysis is somewhat interesting and also seems well thought out. The paper is in general easy to read and understand.

**Weaknesses:**

One major concern is the relationship between the synthetic setup and the real-world instantiation is not well-explained and seems somewhat tenuous to me. From my reading of the paper, it appears that the only major similarity is that both methods involve adding additional tokens to the "fine-tuning" documents (the contextual prompts in the real-world setting and the "original function evaluations" in the synthetic setting. However, these seem structurally different to me: in the synthetic setting, the added context tokens actually depend on the specific inputs and functions used in the example. On the other hand, in the real-world setting, the additional context tokens are sourced from the educational prompts and randomly sampled without considering he contents of a particular example. To summarize, it appears to me that contextual fine-tuning in the synthetic experiments actually provides a significant source of additional supervision (reminiscent of COT approaches) whereas this is absent in real-world instantiation of contextual FT. This makes the connection between the synthetic and real settings somewhat dubious to me and the mechanisms behind the contextual FT remain a bit mysterious.

I also find that the design of the contextual FT prompts used in the real scenario is insufficiently justified. Currently, these prompts are associated with various educational theories. However, to me this appears to be insufficient justification because it is unclear -- and unlikely-- that any parallels can be drawn between the human learning process and the way that large language models use facts. Furthermore, if I understand the paper correctly, it seems that there is no actual task supervision corresponding to these contextual prompts (i.e. for the critical analysis prompt: "Critically analyze the upcoming information. Look for underlying assumptions, evaluate
arguments, and consider different perspectives.”, there is no actual ground-truth supervision given to the model on what the underlying assumptions/arguments in the provided information are). As a result, the mechanisms behind how these contextual prompts actually improve performance are quite unclear, and as I mentioned previously the synthetic data setup is not convincing in its relation to the real data setup.

My concern is further amplified by the insufficient ablation analysis done on the contents of the "contextual prompts". The authors claim that the contents of the prompt are important by their "negative context fine-tuning setup". However, I think that the paper could be further strengthened if they expanded their analysis to non-contradicting context prompts which *are not* inspired by educational theories. This would help me assess the justification of the context prompt design and better understand the source behind the gains.

**Questions:**

1. I would like to see some additional ablations about the contents of the context prompts that go beyond the negative CFT setup. In particular, it would be nice if authors considered prompts consisting of various writing styles (but which are not correlated to educational theories or contradiction of the documents). This could help relate this work to prior research on the personas hypothesis etc.
2. I would like if the authors could further justify how the synthetic data setup should be viewed as comparable to the real-world setup. In particular, why are the additional tokens added in the synthetic settings input dependent while the contextual prompts used in real settings are randomly sampled independently of the contents of the document/example?

---

> ### Author Response · Authors · 2024-11-23
> **Response to Weakness: Synthetic Setting and Question 2**
>
> Thank you very much for your detailed review and insightful comments. We would like to address your concerns.
>
> **Weakness: Synthetic setting and Question 2**
> >In the synthetic setting, the added context tokens actually depend on the specific inputs and functions used in the example. On the other hand, in the real-world setting, the additional context tokens are sourced from the educational prompts and randomly sampled without considering he contents of a particular example. To summarize, it appears to me that contextual fine-tuning in the synthetic experiments actually provides a significant source of additional supervision (reminiscent of COT approaches) whereas this is absent in real-world instantiation of contextual FT.
>
> >I would like it if the authors could further justify how the synthetic data setup should be viewed as comparable to the real-world setup. In particular, why are the additional tokens added in the synthetic settings input dependent while the contextual prompts used in real settings are randomly sampled independently of the contents of the document/example?
>
> Our current hypothesis for why our approach works is that gradients under prompts that contain semantic content relevant for learning serve to regularize the process of learning via fine-tuning. However testing this hypothesis directly is challenging since (a) different LLMs might interpret semantic information in a prompt differently (as a function of scale) and (b) it requires knowing which neurons are responsible for representing the inferred semantic information in the prompt -- an open problem in mechanistic interpretability.
>
> To that end the primary objective of the synthetic experiment was to analyze how contextual prompts affect the gradients of transformer models during training in a controlled setting where we can describe the semantic information that is necessary for learning explicitly via text. The advantage of this is that it enables us to not worry about how the transformer encodes semantic information (thus enabling the study of this phenomenon on much smaller models) and consequently better understand what properties of the gradient enable this.
> To expand on this further, the sequence of tokens we use in the synthetic data, by design, $(x_1,f(x_1),x_2,f(x_2),\ldots,x_k,f(x_k))$ encode the semantic information necessary for learning this synthetic class of problem, facilitated by conditioning on the prompts.
>
> Our empirical results, presented in Appendix G, show that contextual fine-tuning is more effective for instruction-tuned and chat models compared to non-chat models. This observation suggests that models capable of following instructions are better at leveraging contextual prompts during fine-tuning, even when the prompts are not customized to each example. Our intention with the synthetic experiment was to provide insight into the potential mechanisms by which contextual prompts can enhance learning, acknowledging that direct analysis of gradients in large-scale language models is infeasible.

---

> ### Author Response · Authors · 2024-11-23
> **Response to Weakness: Design of the Contextual Prompts**
>
> **Weakness: Design of the contextual prompts**
> >I also find that the design of the contextual FT prompts used in the real scenario is insufficiently justified. Currently, these prompts are associated with various educational theories. However, to me this appears to be insufficient justification because it is unclear -- and unlikely-- that any parallels can be drawn between the human learning process and the way that large language models use facts.
>
> >Furthermore, if I understand the paper correctly, it seems that there is no actual task supervision corresponding to these contextual prompts (i.e. for the critical analysis prompt: "Critically analyze the upcoming information. Look for underlying assumptions, evaluate arguments, and consider different perspectives.”, there is no actual ground-truth supervision given to the model on what the underlying assumptions/arguments in the provided information are). As a result, the mechanisms behind how these contextual prompts actually improve performance are quite unclear, and as I mentioned previously the synthetic data setup is not convincing in its relation to the real data setup.
>
> Thank you for your insightful comments. They indicate we could have been clearer in our manuscript about the hypothesized mechanisms behind how the prompts change the process of learning. There are two core questions you ask, why pick these prompts, and why do these prompts work?
>
> re: your first question -- Our current hypothesis is that there exists prompts whose semantic information can regularize the gradients during learning and improve generalization. This leads to a natural question -- how do you pick these prompts? In general, we do not anticipate the existence of a unique answer to this question and so we chose prompts that were compact yet general purpose. We settled on a variety of prompts summarized from the educational theories because they satisfied our desiderata of being both compact, yet general purpose. We have also experimented with an extension of our approach wherein the prompts are generated by another LLM.  Table 1 in the general comments indicate that such prompts can also enhance model performance, outperforming standard continued pretraining.
>
> What are the implications of this choice and does it imply that LLMs learn using mechanisms similar to humans? We do not believe we have provided sufficient evidence to reach this conclusion.
>
> re: your second question -- do these prompts work? In a nutshell, having a comprehensive understanding of why this phenomena occurs might require a deep dive into the pretraining data, which unfortunately is rarely made available for open source models.
>
> However, we conjecture that the rationale for why our method works has to do with _explanatory text_ that exists in the training corpora.
>
> To ground this conjecture we point the reader to experimental evidence for a different phenomena, chain of thought prompting, A recent work _Why Think Step by Step? Reasoning Emerges from the Locality of Experience (Prystawski et al., 2023) [1]_ suggests that chain of thought prompting works because there are local steps embedded in pretraining corpora that simulate at training time the internal thought process that we have come to expect at test time from CoT prompting. While our goal in this work is to demonstrate the value from CFT, we believe a similar, future study along this lines studying why CFT works, as a function of the pretraining data would be valuable to the community to test this conjecture and shed light on the mechanisms behind why prompts provide useful supervisory signal during learning.

---

> ### Author Response · Authors · 2024-11-23
> **Response to Question 1**
>
> **Question 1: Additional ablations on contextual prompts**
> >I would like to see some additional ablations about the contents of the context prompts that go beyond the negative CFT setup. In particular, it would be nice if authors considered prompts consisting of various writing styles (but which are not correlated to educational theories or contradiction of the documents). This could help relate this work to prior research on the personas hypothesis etc.
>
> Thank you for the suggestion.
>
> To your suggestion of ablations, we have created a variant of our method where instead of CFT with custom education inspired prompts, we experiment with an automated prompt generation system that uses an auxiliary LLM to create prompts for each sampled paragraph. Please see Table 1 in the general comments.
>
> Our findings indicate that models fine-tuned with these automatically generated prompts denoted as AutoDep-CFT show performance improvements over the baseline models without contextual prompts and are comparable to those fine-tuned with our original contextual prompts. This suggests that incorporating prompts with varied writing styles—even those generated automatically without specific alignment to educational theories—can enhance the model's performance.
>
> We are running a few more experiments that we hope can better address this question and will report back soon.
>
> References:
>
> [1] Ben Prystawski and Noah D. Goodman. 2023. Why think step-by-step? Reasoning emerges from the locality of experience. CoRR abs/2304.03843 (2023)

---

> > ### Comment · Reviewer_ULmj · 2024-11-23
> > **Reviewer Response**
> >
> > I would like to thank the authors for their nice rebuttal and clarifying my questions. I believe that it would be very helpful for the authors to incorporate some of what they have written here into their manuscript.
> >
> > Broadly, I think that the authors do showcase an interesting and important finding: the context in fine-tuning examples affects how the model processes the updates. They illustrate this through a synthetic setting as well as real experiments. While the settings are not exactly comparable, I feel that they both support the overall point the authors are trying to make. I also think that the authors make some nice additional experiments to further ablate the role of input-tailored prompts.
> >
> > While I would like to further understand why this phenomena occurs in a realistic setting, I think that the authors findings here are valuable to be shared and thus will be incrementing my score to a 6.

---

### Official Review · Reviewer_fQDe · 2024-11-04

**Soundness:** 2
**Presentation:** 2
**Contribution:** 2
**Rating:** 5
**Confidence:** 3

**Summary:**

The paper presents "Contextual Fine-Tuning" (CFT), a new approach for fine-tuning large language models (LLMs) that incorporates contextual prompts during training. These prompts, designed to mimic cognitive strategies like critical thinking and concept linking, aim to enhance the model's understanding and adaptability in domain-specific tasks, such as finance and medicine.

**Strengths:**

The paper introduces "Contextual Fine-Tuning" (CFT) as an extension of instruction fine-tuning to improve domain-specific learning, which is well-positioned to address limitations in traditional methods.

Extensive experiments demonstrate that CFT improves LLM performance on real-world datasets in domains like finance and medicine. The method yields notable improvements over continued pretraining (CPT) and instruction fine-tuning (IFT).

**Weaknesses:**

I wonder how fine-tuning on a specific domain impacts the language model’s general abilities. The authors could evaluate this by testing the model on general benchmarks to assess if base knowledge and instruction-following abilities are retained or diminished after domain-specific training. This would provide insight into whether contextual fine-tuning maintains the model's versatility across tasks.

**Questions:**

See the weakness part.

---

> ### Author Response · Authors · 2024-11-23
> **Response to Reviewer fQDe**
>
> Thank you to the reviewer for their time and feedback. Please see our response below:
>
> The reviewer recommends evaluating the impact of domain-specific fine-tuning on the model's general and instruction-following capabilities. We evaluate the OpenMedText fine-tuned Llama-2-13B model on general benchmarks and instruction-following benchmarks, and we find that the capabilities of the base model are largely retained. We present results from MMLU, MMLU-Pro, and IFEval which provide coverage of these capabilities.
>
> **Table. Llama-2-13B (Accuracy)**
>
> |      	| Base  | CPT   | CFT   |
> |----------|-------|-------|-------|
> | IFEval   | 0.467 | 0.459 | 0.457 |
> | MMLU 	| 0.478 | 0.483 | 0.479 |
> | MMLU-Pro | 0.187 | 0.165 | 0.164 |
>
>
> We do not observe catastrophic forgetting as a result of fine-tuning, the general knowledge of the model is only slightly diminished. CPT is slightly more robust than CFT to knowledge degradation, however, the performance difference is small and we emphasize CFT's stronger in-domain performance as demonstrated by Table 1 in the manuscript.
>
> **Table. Medical Benchmarks (Manuscript)**
>
> | Llama-2-13B | Anatomy   | Clinical Knowledge | College Biology | College Medicine | Medical Genetics | Professional Medicine | MedQA 	| Average   |
> |-------------|-----------|--------------------|-----------------|------------------|------------------|-----------------------|-----------|-----------|
> | Chat    	| 51.85 	| 56.60          	| 54.17       	| 46.82        	| **63.50**    	| 56.99             	| **45.33** | 53.61 	|
> | Chat (CPT)  | 50.37 	| 60.00          	| 55.90       	| 50.58        	| 62.00        	| 57.35             	| 43.95 	| 54.31 	|
> | Chat (CFT)  | **53.33** | **63.21**      	| **57.99**   	| **56.35**    	| 62.50        	| **57.72**         	| 44.85 	| **56.56** |
>
> To conclude, we find that CFT up-holds a model's general capabilities while providing significant boosts for in-domain performance.
>
> Thank you for your insightful suggestion to evaluate our method on general benchmarks.

---

### Author Response · Authors · 2024-11-23
**General Comment - 1/2**

Dear reviewers,

We sincerely appreciate your effort and valuable feedback. The overall response has been that we develop a flexible, novel and surprising method to improve the knowledge and capabilities of LLMs to improve domain-specific learning and that we present extensive and well-designed experiments to demonstrate that CFT improves language model performance in real-world domains such as finance and medicine.

Several of the reviews have highlighted the the following questions:
1. Reviewer ULmj: Requested more ablations beyond the negative CFT setup, considering prompts with various writing styles not correlated to educational theories or contradictions.
2. Reviewer TQCm: Noted the limited exploration of prompt optimization or automated prompt generation methods.
3. Reviewer eqbM: Suggested using customized contextual prompts that are dependent on the upcoming text.

In response to these suggestions, we have conducted an experiment aimed at addressing these concerns.

**Automated Dependent Contextual Prompt Generation (AutoDep-CFT)**:

We experimented with generating contextual prompts automatically by instructing GPT-4o mini to create prompts based on the content of each batch. This represents a simple alternative of our proposal where instead of the prompts being created via assessments of strategies for human learning, they are generated by another LLM.  Specifically, we used the following instruction template to generate contextual prompts automatically:

"Your task is to create a contextual prompt that guides the LLM's learning process during fine-tuning.

{{ INSTRUCTION }}

{{ MAIN TEXT }}"

In this template:

{{ INSTRUCTION }} is replaced with one of five different instructions derived from our original contextual prompts to generate a variety of prompts. For example:

"Instruction: Given the text below, develop a contextual prompt that leads the reader to compare and contrast the concepts presented with related topics or prior knowledge."
{{ MAIN TEXT }} is replaced with the text from OpenMedText.

By varying the {{ INSTRUCTION }}, we encouraged the model to generate diverse prompts that guide the learning process in different ways.

Below is a list of examples of the contextual prompts generated automatically:

1. "Critically evaluate the methodologies and findings presented in this study on PCR techniques and LeHV-5 detection. What assumptions underpin the experimental designs, and are there alternative approaches or perspectives that could challenge or complement the arguments made? Consider the implications of these methodologies for broader scientific research and diagnostics in veterinary medicine."
2. "Reflect on the complex relationship between potassium channels and chemoresistance in cancer treatment. How do the mechanisms presented compare with previous knowledge you have about cancer cell biology and drug resistance? Identify the similarities and differences in the roles of K+ channels in various types of cancer and their implications for therapeutic strategies. Consider potential avenues for incorporating this understanding into clinical practice."
3. "Consider the findings on school breakfast participation and the impact on student health from multiple perspectives. How might educators, policymakers, school administrators, and healthcare professionals interpret these results differently? Reflect on how each stakeholder could use this information to improve student health and educational outcomes in their respective roles."

---

> ### Author Response · Authors · 2024-11-23
> **General Comment - 2/2**
>
> Results:
>
> **Table 1. Evaluation of CFT with auto-generated contextual prompts that are dependent on the upcoming text on medical benchmarks**
>
> |                	| Anatomy | Clinical Knowledge | College Biology | College Medicine | Medical Genetics | Professional Medicine | MedQA | Average |
> |--------------------|---------|--------------------|-----------------|------------------|------------------|-----------------------|-------|---------|
> | Chat           	| 44.07   | 46.79          	| 48.61       	| 39.02        	| 49.00        	| **48.90**             	| 38.96 | 45.05   |
> | Chat (CPT)     	| 45.19   | 47.17          	| 49.31       	| 43.93        	| 50.50        	| 46.32             	| 39.28 | 45.96   |
> | Chat (CFT)     	| **48.15**   | **48.87**          	| **52.08**       	| 44.22        	| **54.00**        	| 46.69             	| **40.65** | **47.81**   |
> | Chat (AutoDep-CFT) | 45.56   | 48.12          	| 49.31       	| **44.80**        	| 52.50        	| 43.57             	| 40.34 | 46.31   |
>
> **We will refer to this table in more detail in our individual responses to each reviewer.**
>
> The results from Table 1 indicate that AutoDep-CFT, which uses automatically generated, content-dependent prompts, achieves an average accuracy of 46.31%, outperforming both the baseline Chat model (45.05%) and the Chat CPT model (45.96%). While it does not surpass the Chat (CFT) model with manually designed prompts (47.81%), these findings indicate that auto-generated, context-dependent prompts can effectively enhance model performance across medical benchmarks.
>
> We will incorporate the table and update the manuscript. Thank you again for your thoughtful feedback.

---

### Author Response · Authors · 2024-11-28
**Official Comment by Authors**

We thank the reviewers for the valuable feedback. We have made the following changes to address the comments and improve the clarity and robustness of our work.

### Changes to the PDF (indicated in blue for readability, will be modified to black afterwards)

**Main text**
- `[eqbM]` (Section 1) Made minor edits to the first paragraph of the introduction to reduce overstatement.
- (Figure 1) Changed the color scheme from green and red to blue and red to be colorblind-friendly.
- `[ULmj, TQCm]` (Section 3) Provided details on the rationale behind the design of contextual prompts and included a method for generating text-adaptive contextual prompts.
- `[ULmj, eqbM]` (Section 4) Provided a detailed explanation on the settings for the synthetic experiments.
- `[ULmj, TQCm]` (Section 6) Added a summary of the results from the ablation study on text-adaptive contextual prompts and included a summary of the baseline comparison against AdaptLLM.
- `[eqbM]` (Section 7) Removed redundancy and provided more details on future work.

**Appendix**
- `[TQCm]` (Section C) Added the number of journals for each journal category and discussed the limitations of our dataset.
- `[TQCm]` (Section D) Included the computational cost of the training.
- `[ULmj, TQCm]` (Section G.1) Provided details on the ablation study on text-adaptive contextual prompts and presented the results.
- `[TQCm]` (Section H.1) Added the results for a baseline comparison against AdaptLLM.
- `[fQDe]` (Section H.2) Added the results for the evaluation on general and instruction-following benchmarks.

We hope these revisions effectively respond to the reviewers' suggestions and improve the overall quality of the paper. Additionally, we would like to express our sincere gratitude to the reviewers for their insightful suggestions, which have strengthened our manuscript.

---

### Meta-Review · Area_Chair_qTvr · 2024-12-22

**Metareview:**

The authors show that for instruction tuning, prepending a generic prefix to the instruction can improve the performance of the trained model. This is an interesting observation that is simple to implement and could easily become standard practice if the reported performance gains here hold more broadly.

Reviewers generally recommended acceptance, with one borderline exception. While they generally found the paper interesting and the results worth sharing, there were consistent concerns about a general tendency to over-claim in the writing, and especially with the synthetic data section of the paper, which has a tenuous connection with the real-world experiments. The claims about how it is important that the prompts are inspired by cognitive learning theories also do not seem to hold up in light of the new experiments. I strongly encourage the authors to take this feedback into account in the camera-ready and to be more skeptical about their own claims, especially when they rely on anthromorphic generalizations of LLMs. More analysis of what kinds of prompts actually help would be useful.

**Additional Comments On Reviewer Discussion:**

Other than clarifications, most of the questions were around the synthetic data section and about whether it's important that the added prefixes depend on the content of the prompt. The synthetic data section is not particularly important to the paper in its current form, and the authors responded with helpful additional experiments for the latter.

---

### Decision · Program_Chairs · 2025-01-22

Accept (Poster)